# iSHA: Incremental Successive Halving for Hyperparameter Optimization with Budget Constraints

## Abstract

Hyperparameter optimization (HPO) is indispensable for achieving optimal performance in machine learning tasks. While some approaches focus on sampling more promising hyperparameter configurations, methods based on the successive halving algorithm (SHA) focus on efficiently evaluating hyperparameter configurations through the adaptive allocation of evaluation resources and stopping unpromising candidates early. Yet, SHA comes with several hyperparameters itself, one of which is the maximum budget that can be allocated to evaluate a single hyperparameter configuration. Asynchronous extensions of SHA (ASHA) devise a strategy of autonomously increasing the maximum budget and simultaneously allowing for better parallelization. However, while working well in practice with many considered hyperparameter configurations, there are limitations to the soundness of these adaptations when the overall budget for HPO is limited. This paper provides a theoretical analysis of ASHA in applications with budget constraints. We propose incremental SHA (iSHA), a synchronous extension of SHA, allowing to increase the maximum budget. A theoretical and empirical analysis of iSHA shows that soundness is maintained while guaranteeing to be more resource-efficient than SHA. In an extensive set of experiments, we also demonstrate that, in general, iSHA performs superior to ASHA and progressive ASHA.

## 1 Introduction

Hyperparameter optimization (HPO) is a crucial step in the process of engineering machine learning (ML) applications, as optimal performance can only be obtained if parameterized ML algorithms are tuned to the task at hand (Feurer & Hutter, 2019; Bischl et al., 2021). Such a task is specified in the form of a dataset $\mathcal{D}$ and a loss function $\ell$. Typically, HPO is carried out in a trial-and-error fashion by evaluating $\ell$ on the given data $\mathcal{D}$ for various candidate hyperparameter configurations.

In the early days of HPO, grid search and random search (Bergstra et al., 2011) have been the main tools. However, they can be criticized for their disability in finding an optimal hyperparameter configuration as well as their computational cost. In the age of deep learning, a highly efficient HPO method is inevitable, as evaluating hundreds or even thousands of configurations is prohibitive. To address this challenge, several HPO methods have been proposed to improve sampling or evaluation efficiency. For the former, the methods mainly focus on Bayesian Optimization (Hutter et al., 2011), whereas, for the latter, the HPO problem is extended by a budget parameter. Using this parameter, the optimizer can specify for which budget a hyperparameter configuration should be evaluated. This area of the HPO literature is also referred to as multi-fidelity optimization.

Probably the simplest procedure in this area is the Successive Halving Algorithm (SHA), which first evaluates a set of candidates for a minimum starting budget $R_0$, discards the worse half and continues evaluation with the better half for a doubled budget. This procedure is repeated until a maximum budget of $R$ is reached. Thus, concentrating the budget on more promising hyperparameter configurations, the reliability of the evaluations is gradually increased, but also the cost of their evaluations. In contrast, less promising solutions are discarded early on with little budget.

However, for the user of such an HPO method, there are again new hyperparameters to be set: the number of hyperparameter configurations in the starting set $n$, the minimum starting budget $R_0$, and

the maximum budget $R$. Furthermore, in a generalization of SHA, there is an additional reduction parameter $\eta$, which specifies that only $1/\eta$ of the configurations are considered for the next higher budget level. For Hyperband (Li et al., 2018), a heuristic is presented to achieve a satisfactory result with different $n$ and $R_0$, Li et al. (2020) propose an approach that does not need to specify the maximum budget $R$ in advance. This is accomplished by an asynchronous extension of SHA (ASHA), in which decisions about candidate evaluations for larger budgets are made asynchronously, allowing for higher parallelization. This approach has recently been further developed into PASHA (Bohdal et al., 2022) that progressively increases the budget if the ranking of the configurations in the top two high-fidelity sets has not stabilized.

However, asynchronous decision-making comes at the risk of mistakenly promoting hyperparameter configurations to the next budget level. While Li et al. (2020) invoke the law of large numbers to argue that this is not an issue, the problem remains in the case of finite budget constraints, where only a limited number of hyperparameter configurations can be considered.

**Contributions.** To shed light on these limitations, propose potential remedies, and further improve the reliability of these HPO tools, we analyze ASHA and progressive ASHA (PASHA) (Bohdal et al., 2022) from a theoretical and empirical viewpoint. Our contributions can be summarized as follows:

- We provide the first theoretical results for ASHA, analyzing its capabilities in setups with constraints on the overall budget. These findings are accompanied by empirical evidence for a set of HPO benchmarks.
- We propose an incremental (synchronous) extension of SHA (iSHA) that still allows one to increase the maximum allocatable budget $R$ during an HPO run but makes decisions synchronously.
- A theoretical and empirical analysis of iSHA is provided, finding iSHA to be theoretically sound relative to the original SHA, while being provably more resource-efficient.
- In an extensive empirical study, we compare iSHA to the original SHA, and PASHA embedded into the Hyperband framework. We find iSHA to give more robust results compared to PASHA, often yielding higher quality hyperparameter configurations, while being more resource-efficient than SHA.

## 2 HYPERPARAMETER OPTIMIZATION

Hyperparameter optimization (HPO) deals with the problem of finding a suitable parameterization $\boldsymbol{\lambda}$ of an ML algorithm $\mathcal{A}$ with a corresponding hyperparameter space $\Lambda$ for a given task. When facing a typical supervised ML setting, we consider an instance space $\mathcal{X}$ and a target space $\mathcal{Y}$, and elements $\boldsymbol{x} \in \mathcal{X}$ to be (non-deterministically) associated with elements $y \in \mathcal{Y}$ via a joint probability distribution $P$. We assume to be given a (training) dataset $\mathcal{D} = \{\boldsymbol{x}^{(i)}, y^{(i)}\}_{i=1}^{N} \subset \mathcal{X} \times \mathcal{Y}$ from a dataset space $\mathbb{D}$. An ML algorithm $\mathcal{A}$ is a mapping from the dataset space $\mathbb{D}$ and the hyperparameter space $\Lambda$ to a hypothesis space $\mathcal{H} := \{h : \mathcal{X} \rightarrow \mathcal{Y}\} \subseteq \mathcal{Y}^{\mathcal{X}}$, i.e.,

$$A : \mathbb{D} \times \Lambda \rightarrow \mathcal{H}, (\mathcal{D}, \boldsymbol{\lambda}) \mapsto h .$$

The ultimate goal of HPO is to find a parameterization $\boldsymbol{\lambda}^*$ of $\mathcal{A}$, resulting in a hypothesis that minimizes the generalization error (risk $\mathcal{R}$) with respect to some loss function $\ell : \mathcal{Y} \times \mathcal{Y} \rightarrow \mathbb{R}$:

$$\mathcal{R}\big(h = \mathcal{A}(\mathcal{D}, \boldsymbol{\lambda})\big) = \mathbb{E}_{(\boldsymbol{x},y) \sim P}\, \ell(y, h(\boldsymbol{x})) = \int_{(\boldsymbol{x},y) \in \mathcal{X} \times \mathcal{Y}} \ell(y, h(\boldsymbol{x}))\, dP(\boldsymbol{x}, y) .$$

Since the generalization error cannot be computed directly, it is estimated by splitting the data $\mathcal{D}$ into a training and validation set, $\mathcal{D}_{\text{train}}$ and $\mathcal{D}_{\text{val}}$, and computing the validation error:

$$\hat{\boldsymbol{\lambda}} \in \underset{\boldsymbol{\lambda} \in \Lambda}{\arg\min}\, \mathbb{E}_{\mathcal{D}_{\text{train}}, \mathcal{D}_{\text{val}}} \big[\mathbb{E}_{(\boldsymbol{x},y) \in \mathcal{D}_{\text{val}}}\big[\ell(y, \mathcal{A}(\mathcal{D}_{\text{train}}, \boldsymbol{\lambda})(\boldsymbol{x}))\big]\big] .$$

To ease notation, we summarize the expectation as $\hat{\ell}(\boldsymbol{\lambda})$.

As the computation of $\hat{\ell}$ for a $\boldsymbol{\lambda}$ might be costly w.r.t. the available resources (e.g., wall-clock time, number of used data points, etc.), in multi-fidelity HPO, the validation error is usually determined for a certain resource allocation, and thus, its actual value depends on the resources used. Hence, we denote by $\hat{\ell}_r(\boldsymbol{\lambda})$ the validation error of $\mathcal{A}$ with parameterization $\boldsymbol{\lambda}$ and resource allocation $r$. Obviously, the choice of $r$ involves a tradeoff: The more resource units are used, the more accurate the estimate, but the more costly its calculation, and vice versa.

Roughly speaking, a multi-fidelity HPO method seeks to find an appropriate parameterization $\boldsymbol{\lambda}$ of $\mathcal{A}$, while preferably using as few resources as possible, and/or allocating at most a maximum assignable budget $R$ to the evaluation of a hyperparameter configuration $\boldsymbol{\lambda}$ during the search. For convenience, we assume that $R$ is an element of $\mathbb{N} \cup \{\infty\}$, where $R = \infty$ means that resources are not restricted. We define $\hat{\ell}_*(\boldsymbol{\lambda}) := \lim_{r \to R} \hat{\ell}_r(\boldsymbol{\lambda})$ for any $\boldsymbol{\lambda} \in \Lambda$ and $\nu_* := \inf_{\boldsymbol{\lambda} \in \Lambda} \hat{\ell}_*(\boldsymbol{\lambda})$. The goal of an HPO method is then to identify an HPC $\boldsymbol{\lambda}$ belonging to $\arg\min_{\boldsymbol{\lambda} \in \Lambda} \hat{\ell}_*(\boldsymbol{\lambda}) - \nu_*$.

# 3 SUCCESSIVE HALVING AND HYPERBAND

The successive halving algorithm (SHA) (Karnin et al., 2013) solves the non-stochastic best arm identification problem within a fixed budget and was already applied successfully to HPO by Jamieson & Talwalkar (2016a). As a preparation step, a set of $n$ hyperparameter configurations is sampled. Then, starting from a minimum budget $R_0$ for which all the $n$ candidates are evaluated, it iteratively discards the worse half and continues to evaluate the remaining candidates with doubled budget. This procedure is repeated until either only a single hyperparameter configuration is left or a maximum allocatable budget $R$ is reached. Typically, $n$ is chosen such that at least one candidate reaches the final iteration of the algorithm. A budget level for which hyperparameter configurations are evaluated is also referred to as *rung* in the following. Furthermore, we write that a hyperparameter configuration is *promoted* to the next rung if it was not discarded and thus considered in the next iteration of SHA. While SHA allows allocating exponentially more budget on the more promising hyperparameter configurations, its final performance crucially depends on its parameterization. The parameters $n$, $R$ and $R_0$ need to be chosen with care and depending on the task. Starting with too-low an initial budget $R_0$, we face the problem of rejecting actually promising hyperparameter configurations too early, namely those that require more budget, e.g., more data or more training iterations, to perform well enough to remain in the set of promising candidates. In particular, this is typically the case for more complex models, which are known to have more capacity to learn more complex relationships between input space and target space. It is also true for models that are regularized to avoid overfitting. While those models typically need some budget to start to perform well enough, oftentimes, they eventually outperform unregularized candidates resulting in overfitting behavior.

The Hyperband (HB) algorithm (Li et al., 2018) comes with a heuristic of how to choose different values for $n$ and $R_0$, and subsequently uses SHA as a subroutine. This way, different allocation strategies are considered for the tradeoff between (i) considering many configurations $n$ starting with a rather small $R_0$, and (ii) giving some configurations more budget from the beginning. The latter is motivated by the fact that in machine learning, some hyperparameter configurations may require a larger amount of resources to show off their better performance. We refer to each call of SHA as a *bracket* (Li et al., 2018), for which the set of hyperparameters is sampled uniformly at random and given to SHA as an input.

# 4 RELATED WORK

To achieve state-of-the-art performance, hyperparameter optimization (HPO) is an inevitable step in the machine learning process, dealing with finding the most suitable hyperparameter configuration (HPC) of a machine learning algorithm for a given dataset and performance measure. Considering HPO as a black-box optimization problem, various methods can be used to tackle this problem (Feurer & Hutter, 2019; Bischl et al., 2021). Arguably, straightforward solutions include a grid search and a random search. However, both are rather expensive, and thus, methods emerged to improve sample efficiency and evaluation efficiency. While former methods are mostly centered around Bayesian optimization (Frazier, 2018; Hutter et al., 2011), the latter emerged in the branch of multi-fidelity optimization.

In multi-fidelity optimization, the goal is to distribute the budget for evaluating hyperparameter configurations in a way that more budget is concentrated on the more promising configurations and less so on inferior candidates. The Successive Halving algorithm (SHA), initially proposed by Karnin et al. (2013) and later used by Jamieson & Talwalkar (2016b;a) for HPO, devises a powerful HPO method, which has been incorporated as a subroutine in the well-known HPO method `Hyperband` (Li et al., 2018). `Hyperband` has been extended in various directions such as improving its sampling

efficiency (Falkner et al., 2018; Awad et al., 2021; Mallik et al., 2023) and introducing shortcuts in the evaluation process (Mendes et al., 2021).

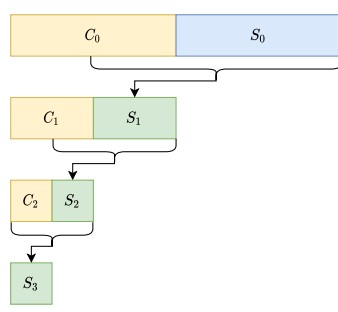

Figure 1: Illustration of how iSHA continues a previously conducted SHA run.

---

**Algorithm 1** Incremental Successive-Halving Algorithm (`iSHA`)

---

**Input:** $S$ set of HPCs, $r$, maximum resource $R$, reduction factor $\eta$, $(C_k)_k$ old sequence of HPCs, $(L_k)_k$ old sequence of losses

**Initialize:** $S_0 \leftarrow S$, $\tilde{n} = |C_0|$, $n = |S_0| + |C_0|$, $s = \log_\eta(R)$

**for** $k \in \{0, 1, \ldots, s\}$ **do**
  $n_k = \lfloor n/\eta^k \rfloor - \lfloor \tilde{n}/\eta^k \rfloor$, $r_k = r\eta^k$
  pull each arm in $S_k$ for $r_k$ times
  **if** $k \leq s - 1$ **then**
    $S_{k+1} \leftarrow$ keep the best $\lfloor n/\eta^{k+1} \rfloor - \lfloor \tilde{n}/\eta^{k+1} \rfloor$ arms from $S_k \cup C_k \backslash C_{k+1}$
  **else**
    $S_{k+1} \leftarrow$ keep the best $\lfloor n/\eta^{k+1} \rfloor$ arms from $S_k \cup C_k$
  **end if**
**end for**

**Output:** Remaining configuration

---

But also SHA has been subject to improvements. In (Li et al., 2020), SHA was extended to asynchronous SHA (ASHA), which helps to better leverage parallel computing resources by promoting candidates asynchronously to the next rung. Simultaneously, the maximum budget $R$ can be adapted on-the-fly. Progressive ASHA (PASHA) proposed by Bohdal et al. (2022) builds on ASHA and incorporates a mechanism to only introduce higher rungs where necessary. While both ASHA and PASHA have been extensively studied empirically, a thorough (theoretical) analysis of the costs of the asynchronous promotion scheme is still lacking. Also, these empirical studies have considered comparably large setups with vast amounts of resources. In our study, we consider small-scale setups and analyze the behavior of ASHA and PASHA in that scope.

## 5 INCREMENTAL SUCCESSIVE HALVING

Due to the static budget setting in SHA, the execution of SHA cannot simply be continued for an adapted parameterization, e.g., a higher maximum allocatable budget $R$. By re-running SHA from scratch, however, knowledge about previously evaluated hyperparameter configurations (HPCs) is discarded and resources already allocated are wasted.

As another extreme, ASHA and PASHA allow to dynamically increase the maximum allocatable budget $R$, devising a scheme for asynchronous promotions to higher rungs. However, as we show in Sections 6 and 7.2, the asynchronous promotions in ASHA and PASHA can be erroneous and thus impede the identification of the optimal hyperparameter configurations.

With incremental successive halving (iSHA), we propose a middle ground for budget-constrained scenarios, i.e., scenarios in which we cannot rely on the law of large numbers as required by Li et al. (2020). Similar to ASHA and PASHA, we allow the maximum allocatable budget to be increased after an SHA run, making SHA in principle stateful. Algorithm 1 translates this into pseudocode. Differences from the original SHA are highlighted in blue. While Algorithm 1 also covers the case of ASHA, adding a single configuration at a time, we assume $|S| + |C_0| = |C_0| \cdot \eta$ for our theoretical and empirical analysis.

In Figure 1 we see the mechanism underlying iSHA to continue a previously conducted run of SHA that resulted in the rungs $C_0, C_1$ and $C_2$. The initially sampled set of HPCs $C_0$ is padded with newly sampled HPCs $S_0$ to initially achieve the same number of HPCs as if SHA had been restarted. However, only the new configurations are executed (following the typical SHA budget allocation) and finally compared with the previous configurations from $C_0$. The already promoted configuration in $C_1$ from the previous SHA run will remain and only the required number of configurations will be promoted, i.e., $S_1$, such that the size of the union of $C_1$ and $S_1$ matches the size of the second rung if SHA had been restarted. This mechanism is then iteratively continued for subsequent rungs.

Intuitively speaking, the strategy of iSHA is to continue a previous SHA run in the most efficient, and thus, resource-saving way. However, similarly to ASHA and PASHA, this efficiency may come at the cost of a potential drop in performance, as previously made decisions cannot be revoked. More specifically, in the worst case, all promotions of the previous run would not have occurred if we had known the complete set of candidate HPCs from the start. Only filling up the rungs leaves less space for the desired candidates to be promoted to the highest rung.

Nevertheless, we prove in the next section that we are still able to identify near-optimal solutions with a high probability, which will be confirmed by empirical results in Section 7.2 later on. Furthermore, we demonstrate that this robustness gives iSHA an edge over ASHA and PASHA when it comes to the quality of returned hyperparameter configurations in settings with limited budget.

## 6 THEORETICAL RESULTS

We split the theoretical results into three parts. First, we provide a theoretical analysis of ASHA. Second, we give some theoretical guarantees for iSHA, our extension of SHA, and third, we extend these guarantees to an incremental extension of Hyperband. Since the Successive Halving algorithm solves a multi-armed bandit problem (Jamieson & Talwalkar, 2016b), in the following analysis, we will stick to the notation and terms of multi-armed bandits. Multi-arm bandit problems are transferred to HPO by (a) considering a hyperparameter configuration $\boldsymbol{\lambda}$ as an arm $i$ and (b) when drawing an arm $i$ for $k$ times, observing the loss $\ell_{i,k}$ (compare to Section 2).

In the same spirit as in (Jamieson & Talwalkar, 2016b; Li et al., 2018), we need the following assumption as a prerequisite for the theoretical analyses of the next subsections.

**Assumption 6.1.** For each arm $i \in \mathbb{N}$, the limit $\nu_i := \lim_{t \to \infty} \ell_{i,t}$ exists.

Moreover, we denote the convergence speed by $\gamma(t) \geq \sup_i |\ell_{i,t} - \nu_i|, \forall t \in \mathbb{N}$.

### 6.1 THEORETICAL ANALYSIS OF ASHA

We now analyze ASHA (Li et al., 2020), which, to the best of our knowledge, is the only algorithm with a similar goal of more efficient resource use as our proposed incremental SH variant.

**Theorem 6.2** (Necessary Budget for ASHA)**.** *Fix $n$ arms and assume $\nu_1 \leq \ldots \leq \nu_n$. Let*

$$z_{\text{ASHA}} = (\lfloor \log_\eta(n) \rfloor + 1) \cdot n \cdot \max \Big\{ \max_{k \in [K]} \eta^{-k} \gamma^{-1} \big( \tfrac{\nu_{\lfloor \text{rung}_{k-1} \rfloor / \eta \rfloor + 1} - \nu_1}{2} \big),$$
$$\eta^{-K} \max_{i \in \text{rung}_K \setminus \{1\}} \gamma^{-1} \big( \tfrac{\nu_i - \nu_1}{2} \big) \Big\},$$

*where $K \leq \lfloor \log_\eta(n) \rfloor$ is the top rung of ASHA. If ASHA is run with some budget $B \geq z_{\text{ASHA}}$, then the best arm $1$ is returned.*

The dependence is linear-logarithmic in $n$, and the limit gap from the best arm to the other arms occurs in the inverted convergence rate $\gamma^{-1}$. The first term in the maximum makes sure that the best arm reaches the top rung $K$, while the second term makes sure that the best arm will eventually be returned. As a corollary of the proof of Theorem 6.2 (see Section C), we obtain the following result.

**Corollary 6.3** (Worst Case Promotion Costs)**.** *Assume all rungs to be full, i.e., no promotion is possible, and the top rung $K$ only contains the current incumbent arm. If at that time a new best arm (configuration) $\hat{i}$ is sampled, then promoting $\hat{i}$ to the sole solution of the new top rung $K + 1$ requires the sampling of $\eta^K - 1$ additional arms (HPCs) and a total of $\eta^{K+1}$ many jobs.*

From these results, we can draw two major conclusions. The more configurations have already been considered in ASHA when $\hat{i}$ enters the pool of considered hyperparameter configurations, i.e., the later in the process, the more budget needs to be spent to promote $\hat{i}$ to the top rung. Particularly, in a scenario with a limited budget, e.g., limited by the overall budget or by the number of configurations to be sampled, ASHA fails to return $\hat{i}$, if the required budget for promoting the best configuration exceeds the remaining budget. A similar result can be shown for PASHA.

## 6.2 THEORETICAL ANALYSIS OF INCREMENTAL SUCCESSIVE HALVING

For iSHA (Algorithm 1), we first prove a lower bound on the budget to return a nearly optimal arm (configuration). The proof is given in Appendix B.1.

**Theorem 6.4** (Necessary Budget for iSHA). *Fix $n$ arms from which $\tilde{n}$ arms were already promoted, and assume $\nu_1 \leq \cdots \leq \nu_n$. For any $\epsilon > 0$ let*

$$z_{\text{iSHA}} = \eta \lceil \log_\eta(n) \rceil \cdot \max_{i=2,\ldots,n} i \left(1 + \min\left\{R, \gamma^{-1}\left(\max\left\{\tfrac{\epsilon}{4}, \tfrac{\nu_i - \nu_1}{2}\right\}\right)\right\}\right).$$

*If any of the iSHA is run with some budget $B \geq z_{\text{C−SH}}$, then an arm $\hat{i}$ is returned that satisfies $\nu_{\hat{i}} - \nu_1 \leq \epsilon/2$.*

Further, we can specify the improvement of iSHA over the costly re-run of SH.

**Theorem 6.5** (Improvement of number of pulls of iSHA in comparison to SHA). *Fix $n$ arms, a maximal size of $R$, $r$ and $\eta$. Assume that we have already run SHA on $\tilde{n}$ arms and the same values for $B$, $r$, and $\eta$. Let $\eta_- = \eta - 1$ and $s^+ = s + 1$. If we ran SHA, iSHA over $s$ rounds with the above variables, we have*

$$\frac{\#\{\text{total pulls of } \text{iSHA}\}}{\#\{\text{total pulls of SH}\}} \leq 1 - \frac{(s^+)(\tilde{n}R + \eta^s)(\eta_-) - (\eta^{s^+} - 1)(2R + n)}{(s^+)(nR + \eta^s)(\eta_-) - (\eta^{s^+} - 1)(R + n)}.$$

Again, as a corollary of the proof of Theorem 6.5 (see Section Appendix B.2), we obtain the following result regarding the "limit case", i.e., if we would increase the maximum size $R$ infinitely often, or, equivalently, the number of possible rungs $s$ infinitely often.

**Corollary 6.6.** *If we run iSHA and SHA infinitely many times with*

*(i) an ever-increasing maximum size $R$, and*
*(ii) such that the newly sampled number of configurations in each new run of iSHA fulfills $|S| + |C_0| = |C_0| \cdot \eta$, where $C_0$ is the number of configurations in the previous run,*

*then the ratio of total pulls of iSHA and total pulls of SHA converges to $1 - \eta^{-1}$.*

Note that a comparison similar to Theorem 6.5 is difficult to make, since ASHA does not include the parameter $R$.

## 6.3 INCREMENTAL-HYPERBAND

Like the original version of SHA and its extensions ASHA and PASHA, we can also employ iSHA as a subroutine in Hyperband. To this end, Hyperband needs to be made incremental in itself, as done in Algorithm 2 in the appendix, which we call iHB (incremental Hyperband). In the following, we provide a theoretical analysis of this incremental version of Hyperband with iSHA as a subroutine. Figure 2 illustrates how every Hyperband bracket is updated after increasing the maximum budget $R$.

An optimal hyperparameter configuration $\boldsymbol{\lambda}^*$ as defined above may not always exist. Even if it does, it could be infeasible to search for it as our hyperparameter configuration space is usually very large or even infinite. Therefore, we will relax our goal and seek to find a configuration that is at least "nearly optimal", akin to the literature on HPO problems: For $\epsilon > 0$, we call $\hat{\boldsymbol{\lambda}}$ an $\epsilon$-*optimal configuration* iff $\nu_{\hat{\boldsymbol{\lambda}}} - \nu_{\boldsymbol{\lambda}^*} \leq \epsilon$. To ensure that the search for such a configuration does not resemble the search for a needle in a haystack, we need an assumption which guarantees that the probability of the existence of an $\epsilon$-optimal configuration in our sampled set of configurations is high enough.

**Assumption 6.7.** The proportion of $\epsilon$-optimal configurations in $\Lambda$ is $\alpha \in (0,1)$.

Note that we now have at least one $\epsilon$-optimal configuration in a sampled set of configurations with probability at least $1 - \delta$, if the sample size is at least $\lceil \log_{1-\alpha}(\delta) \rceil$ for a fixed failure probability $\delta \in (0,1)$. With this, we can state the following theorem, the proof of which is given in Appendix B.3.

**Theorem 6.8.** *Let $\eta, R, \alpha$ and $\delta$ be fixed such that*

$$R \geq \max\left\{ \lceil \log_{1-\alpha}(\delta) \rceil (\eta_-) + 1, \eta \bar{\gamma}^{-1}\left(L_{\eta, L_{\eta,R}} + 4 + \frac{\lfloor L_{\eta,R} \rfloor}{2} - \frac{\sum_{k=1}^{\lfloor L_{\eta,R} \rfloor + 1} \log_\eta(k)}{\lfloor L_{\eta,R} \rfloor + 1}\right)\right\}$$

*for $\bar{\gamma}^{-1} := \max_{s=0,\ldots,\lfloor L_{\eta,R} \rfloor} \max_{i=2,\ldots,n_s} i\left(1 + \min\left\{R, \gamma^{-1}\left(\max\left\{\tfrac{\epsilon}{4}, \tfrac{\nu_i - \nu_1}{2}\right\}\right)\right\}\right)$*

*and $L_{\eta,R} = \log_\eta(R)$, then iHB finds an $\epsilon$-optimal configuration with probability at least $1 - \delta$.*

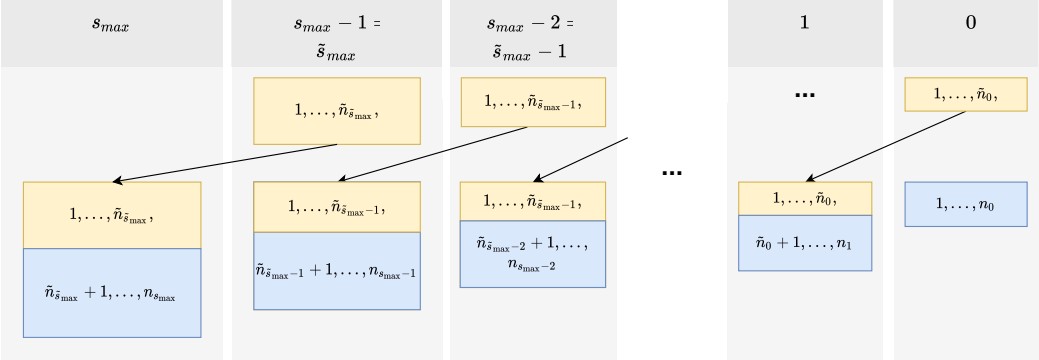

Figure 2: Illustration of how the brackets of incremental hyperband are arranged and filled up when the maximum budget $R$ is increased.

To conclude, despite the incremental extension of Hyperband, we can maintain the theoretical guarantees of the original Hyperband. Although promotions in iSHA are also to some extent performed asynchronously, we can still identify the best arm when doing promotions in a batch, provided a sufficiently large batch size.

## 7  EMPIRICAL EVALUATION

In addition to the theoretical results of the previous section, we evaluate iSHA empirically and compare it to PASHA (Bohdal et al., 2022), and SHA (Jamieson & Talwalkar, 2016a).

We are especially interested in the following two research questions:

**RQ1**  Is iSHA able to retain the quality of returned HPCs as compared to applying SHA from scratch?
**RQ2**  How does the proposed iSHA compare to the state-of-the-art algorithm PASHA?

### 7.1  EXPERIMENT SETUP

In our experimental evaluation, we compare iSHA, to PASHA, and SHA as subroutines embedded in Hyperband to answer the research questions **RQ1** and **RQ2**. Note that we do not include ASHA as it was demonstrated to perform inferior to PASHA in (Bohdal et al., 2022). To this end, we conduct an extensive set of experiments tackling various HPO tasks, considering various types of learners and two different fidelity parameters: the number of *epochs* and the *fraction* of the training data used for fitting a model.

As a benchmark library, we use YAHPO Gym (Pfisterer et al., 2022), which provides fast-to-evaluate surrogate benchmarks for HPO with particular support for multi-fidelity optimization, rendering it a perfect fit for our study. From YAHPO Gym, we select the benchmarks listed in Table 1. All the benchmarks consist of several datasets, which are referred to as benchmark instances, allowing for a broad comparison. Due to space limitations, we only present a summary of the results here, whereas detailed results can be found in Appendix D.

Furthermore, we set the initial max size $R_{t-1} = 16$ and increase it after the first run by a factor of $\eta$ to $R_t = \eta R_{t-1}$, as this is a budget that is supported by all benchmark scenarios. Since ASHA and PASHA automatically increase the maximum budget depending on the observed performances, we only ensure an upper limit of $R_t$ for both to ensure a fair comparison. As a termination criterion, we use that the number of HPCs would exceed the pool size of the Hyperband bracket. For benchmarks considering a fraction of the training dataset as fidelity parameter, we translate a budget $r$ by $r/R_t$ into a fraction between 0 and 1.

Furthermore, we repeat each combination of algorithm, $\eta$, and benchmark instance for 30 seeds resulting in a total amount of $30 \times 3 \times 2 \times 378 = 68,040$ hyperparameter optimization runs. We computed all experiments on a single workstation equipped with 2xIntel Xeon Gold 5122 and 256GB RAM. The code and data will be made publicly available via GitHub.

Table 1: List of considered benchmarks from YAHPO-Gym with the type of learner, number of considered datasets, objective function, and the type of budget that can be used as a fidelity parameter.

| Benchmark | Model | # Inst. | Objective | Fidelity |
|---|---|---|---|---|
| lcbench | neural network | 34 | val_accuracy | epochs |
| rbv2_svm | SVM | 106 | acc | fraction |
| rbv2_ranger | random forest | 119 | acc | fraction |
| rbv2_xgboost | XGBoost | 119 | acc | fraction |

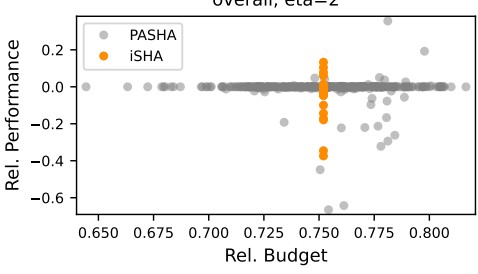 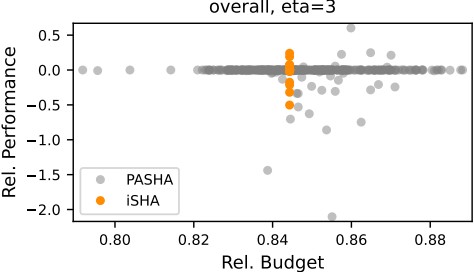

Figure 3: Scatter plots relating the performance on the $y$-axis and the consumed budget on the $x$-axis to the performance achieved and budget consumed by SHA. Note that the ranges for the performance and budget vary from $\eta = 2$ (left) to $\eta = 3$ (right). For the relative budget lower is better whereas for relative performance higher is better.

## 7.2 EMPIRICAL RESULTS

In Figure 3 we present the performance of the finally chosen hyperparameter configuration and the budget spent by the Hyperband optimizer in relation to the performance of the solution returned and the budget consumed by SHA. Hence, a relative performance of $0.0$ means that the solution quality matches the one returned by SHA, a larger (smaller) value an improvement (degradation) w.r.t. SHA. The relative budget denotes the share of the budget that SHA consumes by re-running from scratch. Therefore, a relative budget of 1 means that the consumed budget is on par with the budget of SHA. A lower relative budget correspondingly means that less budget was consumed.

As can be seen, our iSHA extension robustly yields competitive performance to re-running SHA from scratch for a larger maximum assignable budget $R$, while substantially reducing the consumed budget to roughly 75% for $\eta = 2$ and $84.5\%$ for $\eta = 85\%$. Regarding **RQ1**, we can confirm that iSHA retains the quality of returned HPCs.

On the contrary, the performance and budget consumption of PASHA shows a lot of variance, including variations in all possible directions: improvements and degradations in performance, using more or less budget than iSHA. Since higher rungs are only introduced in PASHA whenever necessary, i.e., if the soft ranking over the configurations of the last two rungs changes, PASHA has the potential to reduce the consumed budget even more than iSHA does. However, there is no guarantee that PASHA will use less budget than iSHA, and also in terms of performance, PASHA is clearly less robust.

Table 2: Aggregated statistics across benchmark instances comparing the performance and budget to natively applying SHA. Differences in accuracy larger than 0.001 are considered for improvements or degradations. (left: $\eta = 2$, right: $\eta = 3$)

| $\eta = 2$ | Performance | | | Budget | |
|---|---|---|---|---|---|
| Approach | Impr | Degr | Tie | Mean | Std |
| PASHA | 21 | 69 | 288 | 0.7521 | 0.0274 |
| iSHA | 7 | 14 | 357 | 0.7520 | 0.0 |

| $\eta = 3$ | Performance | | | Budget | |
|---|---|---|---|---|---|
| Approach | Impr | Degr | Tie | Mean | Std |
| PASHA | 24 | 88 | 266 | 0.8483 | 0.0145 |
| iSHA | 9 | 14 | 355 | 0.8443 | 0.0 |

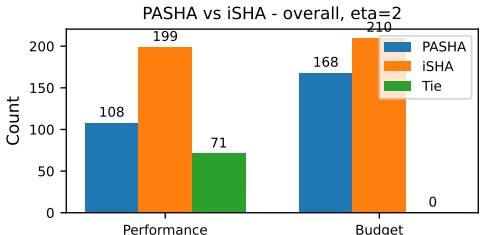 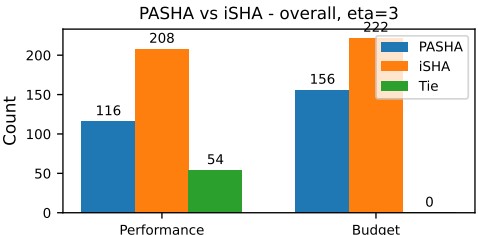

Figure 4: Bar charts counting the number of datasets for which either PASHA or iSHA performed best with respect to performance and accumulated budget for evaluating hyperparameter configurations. The plots show the results for $\eta = 2$ on the left and $\eta = 3$ on the right.

This is again confirmed by the results in Table 2, where we simply count the number of benchmark instances for which an improvement, degradation, or tie w.r.t. the performance of SHA is obtained. While PASHA gives the most improvements in terms of performance for both values of $\eta$, it also comes with the most performance degradations which even outnumber the improvements by a factor of 3 to 4. Furthermore, we provide the average and the standard deviations for the relative budget consumed across the benchmark instances. On average, for both values of $\eta$, iSHA consumes the least budget, whereas, for PASHA, the standard deviation is an order of magnitude larger compared to the other approaches.

In a direct comparison between PASHA and iSHA, we compare the quality of the finally returned hyperparameter configurations to each other in Figure 4. Furthermore, we compare the accumulated budget of the two approaches and count how many times either of the two performed better than the other one. For the performance comparison, we only consider differences in performance larger than 0.001 as better performance. From the plots we can see that on the majority of the datasets iSHA performs better than PASHA for both performance and accumulated budget and independent of the choice for $\eta$. However, the savings in the budget are a little more pronounced in the case of $\eta = 3$.

From these results, we can conclude that iSHA is a robust and more resource-efficient incremental version of SHA, and the theoretical guarantees given in the previous section can be validated in practice as well. PASHA is able to reduce the consumed budget drastically, and its heuristic nature may allow one to achieve substantial reductions in budget, albeit these improvements are not obtained consistently. Overall, we find iSHA to perform way more robust in general, and as detailed in the appendix, compare favorably in a direct comparison to PASHA.

## 8 Conclusion and Future Work

In this paper, we proposed an extension to the well-known HPO method Successive Halving (SHA), called Incremental Successive Halving (iSHA), aiming to improve its efficiency when the max size hyperparameter $R$ of SHA needs to be increased post-hoc. We derived theoretical guarantees on the quality of the final choice, as well as on the saved budget, when a previous SHA run is continued. Furthermore, we provide the first theoretical analysis of asynchronous SHA, emphasizing the price that needs to be paid for the asynchronous promotions. In an empirical study, we also find that iSHA yields results similar to the much more expensive baseline variant of SHA and often better results than the current state-of-art among the asynchronous variants of SHA. In fact, our approach only requires the budget of the sole run with the increased max size.

In future work, we plan to combine our SHA extensions with more sophisticated strategies for sampling hyperparameter configurations, as for example done by Awad et al. (2021) or Falkner et al. (2018) and HyperJump, to improve iHB's efficacy and efficiency even further. Another interesting avenue of future research is outlined by PriorBand where a prior distribution is incorporated for sampling new hyperparameter configurations (Mallik et al., 2023).

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
