\big( 1 + \min \big\{ R, \gamma^{-1} \big( \max \big\{ \tfrac{\epsilon}{4}, \tfrac{\nu_i - \nu_1}{2} \big\} \big) \big\} \big).$$

*If any of the iSHA is run with some budget $B \geq z_{\text{C-SH}}$, then an arm $\hat{i}$ is returned that satisfies $\nu_{\hat{i}} - \nu_1 \leq \epsilon/2$.*

Further, we can specify the improvement of iSHA over the costly re-run of SH.

**Theorem 6.5** (Improvement of number of pulls of iSHA in comparison to SHA). *Fix $n$ arms, a maximal size of $R$, $r$ and $\eta$. Assume that we have already run SHA on $\tilde{n}$ arms and the same values for $B$, $r$, and $\eta$. Let $\eta_- = \eta - 1$ and $s^+ = s + 1$. If we ran SHA, iSHA over $s$ rounds with the above variables, we have*

$$\frac{\#\{\text{total pulls of } \text{iSHA}\}}{\#\{\text{total pulls of SH}\}} \leq 1 - \frac{(s^+)(\tilde{n}R + \eta^s)(\eta_-) - (\eta^{s^+} - 1)(2R + n)}{(s^+)(nR + \eta^s)(\eta_-) - (\eta^{s^+} - 1)(R + n)}.$$

Again, as a corollary of the proof of Theorem 6.5 (see Section Appendix B.2), we obtain the following result regarding the "limit case", i.e., if we would increase the maximum size $R$ infinitely often, or, equivalently, the number of possible rungs $s$ infinitely often.

**Corollary 6.6.** *If we run iSHA and SHA infinitely many times with*

  *(i) an ever-increasing maximum size $R$, and*
  *(ii) such that the newly sampled number of configurations in each new run of iSHA fulfills $|S| + |C_0| = |C_0| \cdot \eta$, where $C_0$ is the number of configurations in the previous run,*

*then the ratio of total pulls of iSHA and total pulls of SHA converges to $1 - \eta^{-1}$.*

Note that a comparison similar to Theorem 6.5 is difficult to make, since ASHA does not include the parameter $R$.

## 6.3 Incremental-Hyperband

Like the original version of SHA and its extensions ASHA and PASHA, we can also employ iSHA as a subroutine in Hyperband. To this end, Hyperband needs to be made incremental in itself, as done in Algorithm 2 in the appendix, which we call iHB (incremental Hyperband). In the following, we provide a theoretical analysis of this incremental version of Hyperband with iSHA as a subroutine. Figure 2 illustrates how every Hyperband bracket is updated after increasing the maximum budget $R$.

An optimal hyperparameter configuration $\boldsymbol{\lambda}^*$ as defined above may not always exist. Even if it does, it could be infeasible to search for it as our hyperparameter configuration space is usually very large or even infinite. Therefore, we will relax our goal and seek to find a configuration that is at least "nearly optimal", akin to the literature on HPO problems: For $\epsilon > 0$, we call $\hat{\boldsymbol{\lambda}}$ an *$\epsilon$-optimal configuration* iff $\nu_{\hat{\boldsymbol{\lambda}}} - \nu_{\boldsymbol{\lambda}^*} \leq \epsilon$. To ensure that the search for such a configuration does not resemble the search for a needle in a haystack, we need an assumption which guarantees that the probability of the existence of an $\epsilon$-optimal configuration in our sampled set of configurations is high enough.

**Assumption 6.7.** The proportion of $\epsilon$-optimal configurations in $\Lambda$ is $\alpha \in (0, 1)$.

Note that we now have at least one $\epsilon$-optimal configuration in a sampled set of configurations with probability at least $1 - \delta$, if the sample size is at least $\lceil \log_{1-\alpha}(\delta) \rceil$ for a fixed failure probability $\delta \in (0, 1)$. With this, we can state the following theorem, the proof of which is given in Appendix B.3.

**Theorem 6.8.** *Let $\eta, R, \alpha$ and $\delta$ be fixed such that*

$$R \geq \max \bigg\{ \lceil \log_{1-\alpha}(\delta) \rceil (\eta_-) + 1, \eta \bar{\gamma}^{-1} \Big( L_{\eta, L_{\eta,R}} + 4 + \frac{\lfloor L_{\eta,R} \rfloor}{2} - \frac{\sum_{k=1}^{\lfloor L_{\eta,R} \rfloor + 1} \log_\eta(k)}{\lfloor L_{\eta,R} \rfloor + 1} \Big) \bigg\}$$

*for $\bar{\gamma}^{-1} := \max_{s=0,\ldots,\lfloor L_{\eta,R} \rfloor} \max_{i=2,\ldots,n_s} i \big( 1 + \min \big\{ R, \gamma^{-1} \big( \max \big\{ \tfrac{\epsilon}{4}, \tfrac{\nu_i - \nu_1}{2} \big\} \big) \big\} \big)$*

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

|---|---|---|---|---|---|
| PASHA | 21 | 69 | 288 | 0.7521 | 0.0274 |
| iSHA | 7 | 14 | 357 | 0.7520 | 0.0 |

| $\eta = 3$ Approach | Performance Impr | Degr | Tie | Budget Mean | Std |
|---|---|---|---|---|---|
| PASHA | 24 | 88 | 266 | 0.8483 | 0.0145 |
| iSHA | 9 | 14 | 355 | 0.8443 | 0.0 |

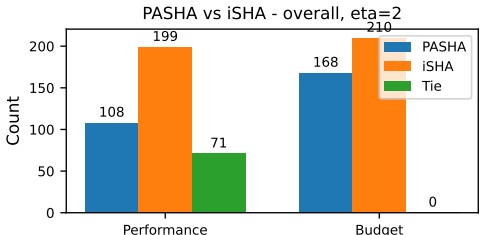 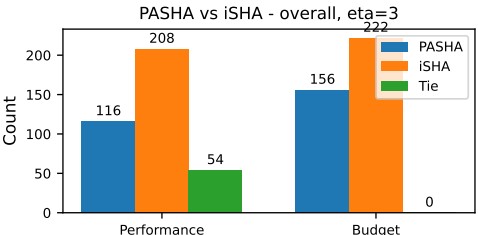

Figure 4: Bar charts counting the number of datasets for which either PASHA or iSHA performed best with respect to performance and accumulated budget for evaluating hyperparameter configurations. The plots show the results for $\eta = 2$ on the left and $\eta = 3$ on the right.

This is again confirmed by the results in Table 2, where we simply count the number of benchmark instances for which an improvement, degradation, or tie w.r.t. the performance of SHA is obtained. While PASHA gives the most improvements in terms of performance for both values of $\eta$, it also comes with the most performance degradations which even outnumber the improvements by a factor of 3 to 4. Furthermore, we provide the average and the standard deviations for the relative budget consumed across the benchmark instances. On average, for both values of $\eta$, iSHA consumes the least budget, whereas, for PASHA, the standard deviation is an order of magnitude larger compared to the other approaches.

In a direct comparison between PASHA and iSHA, we compare the quality of the finally returned hyperparameter configurations to each other in Figure 4. Furthermore, we compare the accumulated budget of the two approaches and count how many times either of the two performed better than the other one. For the performance comparison, we only consider differences in performance larger than 0.001 as better performance. From the plots we can see that on the majority of the datasets iSHA performs better than PASHA for both performance and accumulated budget and independent of the choice for $\eta$. However, the savings in the budget are a little more pronounced in the case of $\eta = 3$.

From these results, we can conclude that iSHA is a robust and more resource-efficient incremental version of SHA, and the theoretical guarantees given in the previous section can be validated in practice as well. PASHA is able to reduce the consumed budget drastically, and its heuristic nature may allow one to achieve substantial reductions in budget, albeit these improvements are not obtained consistently. Overall, we find iSHA to perform way more robust in general, and as detailed in the appendix, compare favorably in a direct comparison to PASHA.

## 8 CONCLUSION AND FUTURE WORK

In this paper, we proposed an extension to the well-known HPO method Successive Halving (SHA), called Incremental Successive Halving (iSHA), aiming to improve its efficiency when the max size hyperparameter $R$ of SHA needs to be increased post-hoc. We derived theoretical guarantees on the quality of the final choice, as well as on the saved budget, when a previous SHA run is continued. Furthermore, we provide the first theoretical analysis of asynchronous SHA, emphasizing the price that needs to be paid for the asynchronous promotions. In an empirical study, we also find that iSHA yields results similar to the much more expensive baseline variant of SHA and often better results than the current state-of-art among the asynchronous variants of SHA. In fact, our approach only requires the budget of the sole run with the increased max size.

In future work, we plan to combine our SHA extensions with more sophisticated strategies for sampling hyperparameter configurations, as for example done by Awad et al. (2021) or Falkner et al. (2018) and HyperJump, to improve iHB's efficacy and efficiency even further. Another interesting avenue of future research is outlined by PriorBand where a prior distribution is incorporated for sampling new hyperparameter configurations (Mallik et al., 2023).

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

## ORGANIZATION OF APPENDIX

## A  PSEUDO-CODES

**Incremental Hyperband.** The incremental Hyperband variant mentioned in Section 6.3 is given in Algorithm 2, where all differences to the original `Hyperband` algorithm by Li et al. (2018) are indicated by a blue text color.

**Variants of iSHA.** While iSHA is arguably the most efficient way to continue a previous run of SHA, there are also other possible ways to do so. One way, which we call discarding Incremental-SuccessiveHalving (given in Algorithm 3), is when the start pool of hyperparameter configurations is extended by the new hyperparameter configurations, it is allowed to discard hyperparameter configurations that were promoted in the previous run and have already been evaluated on a larger budget. Another way that is more efficient and reuses previous evaluations of hyperparameter configurations, is by conserving the information about hyperparameter configurations that have already been evaluated for a specific budget but have been discarded in a previous iteration. In this way, hyperparameter configurations that were already discarded are allowed to return to the pool of promising candidates. This variant will be called preserving Incremental-SuccessiveHalving algorithms and is given in Algorithm 4.

---

**Algorithm 2** Incremental-Hyperband (iHB)

---

**Input:** max size $R$, $\eta \geq 2$, old max size $\tilde{R} \in \{0, R/\eta\}$, old sequence of configuration samples $((C_{s,k})_{k\in\{0,...,s\}})_{s\in\{0,...,\log_\eta(\tilde{R})\}}$ and losses $((L_{s,k})_{k\in\{0,...,s\}})_{s\in\{0,...,\log_\eta(\tilde{R})\}}$

**Initialize:** $s_{max} = \lfloor \log_\eta(R) \rfloor$, $B = (s_{max} + 1)R$

**if** $\tilde{R} > 0$ **then**
    $\tilde{s}_{max} = \lfloor \log_\eta(\tilde{R}) \rfloor = s_{max} - 1$, $\tilde{B} = (\tilde{s}_{max} + 1)\tilde{R}$
**end if**

**for** $s \in \{s_{max}, s_{max} - 1, \ldots, 0\}$ **do**
    $n_s = \lceil \frac{B}{R} \frac{\eta^s}{(s+1)} \rceil$, $r_s = R/\eta^s$
    **if** $\tilde{R} > 0$ and $s > 0$ **then**
        $\tilde{s} = s - 1$, $\tilde{n}_s = \lceil \frac{\tilde{B}}{\tilde{R}} \frac{\eta^{\tilde{s}}}{(\tilde{s}+1)} \rceil$, $\tilde{r}_s = \tilde{R}/\eta^{\tilde{s}} = r_s$
    **else**
        $\tilde{n}_s = 0$
    **end if**
    $S \leftarrow$ sample $n_s - \tilde{n}_s$ configurations
    xID-SuccessiveHalving($S, B, r_s, R, \eta, (C_{\tilde{s},k})_{k\in\{0,...,\tilde{s}\}}, (L_{\tilde{s},k})_{k\in\{0,...,\tilde{s}\}}$)
**end for**

**Output:** Configuration with smallest intermediate loss

---

**Algorithm 3** Discarding Incremental-SuccessiveHalving (`d-iSHA`)

---

**Input:** $S$ set of arms, budget $B$, $r$, max size $R$, $\eta$, $(C_k)_k$ old sequence of configurations, $(L_k)_k$ old sequence of losses

**Initialize:** $S_0 \leftarrow S \cup C_0$, $n = |S_0|$

**for** $k \in \{0, 1, \ldots, s\}$ **do**
    $n_k = \lfloor n/\eta^k \rfloor$, $r_k = r\eta^k$
    pull each arm in $S_k \backslash C_k$ for $r_k$ times
    $S_{k+1} \leftarrow$ keep the best $\lfloor n/\eta^{k+1} \rfloor$ arms from $S_k$
**end for**

**Output:** Remaining configuration

---

**Algorithm 4** Preserving Incremental-SuccessiveHalving (`p-iSHA`)

---

**Input:** $S$ set of arms, budget $B$, $r$, max size $R$, $\eta$, $(C_k)_k$ old sequence of configurations, $(L_k)_k$ old sequence of losses

**Initialize:** $S_0 \leftarrow S \cup C_0$, $n = |S_0|$

**for** $k \in \{0, 1, \ldots, s\}$ **do**
    $n_k = \lfloor n/\eta^k \rfloor$, $r_k = r\eta^k$
    pull each arm in $S_k \backslash C_k$ for $r_k$ times
    $S_{k+1} \leftarrow$ keep the best $\lfloor n/\eta^{k+1} \rfloor$ arms from $S_k \cup C_k$
**end for**

**Output:** Remaining configuration

---

# B PROOFS

## B.1 PROOF OF THEOREM 6.4

*Proof of Theorem 6.4.* This proof consists of two parts: First, we will focus on the efficient Incremental-SuccessiveHalving Algorithm given in Algorithm 1. Second, we will show a similar lower bound on the number of necessary samples for the discarding and preserving Incremental-SuccessiveHalving algorithms given in Algorithm 3 and Algorithm 4.

**Part I: iSHA analysis**

Step 1: Algorithm 1 never exceeds the budget $B$, which can be seen as follow. The budget used is bounded by

$$
\begin{aligned}
\sum_{k=0}^{s} n_k r_k &= \sum_{k=0}^{s} \left( \lfloor n/\eta^k \rfloor - \lfloor \tilde{n}/\eta^k \rfloor \right) \frac{R\eta^k}{\eta^s} \\
&\leq \sum_{k=0}^{s} \left( \lfloor n/\eta^k \rfloor \right) \frac{R\eta^k}{\eta^s} \\
&\leq \sum_{k=0}^{s} \frac{(s_{\max}+1)\eta^s}{(s+1)} \frac{R}{\eta^s} \\
&\leq (s_{\max}+1)R = B.
\end{aligned}
$$

Step 2: Let $n_k = |S_k| + |C_k|$ and $\tilde{n}_k = |C_k|$ such that $n_0 = n$ and $\tilde{n}_o = \tilde{n}$. Without loss of generality, we assume that the limit values of the losses are ordered, such that $\nu_1 \leq \nu_2 \leq \cdots \leq \nu_n$. Note, that due to the above condition also the limit values of arms in $S_k$ and resp. in $C_k$ are ordered, e.g. for $\nu_i, \nu_j \in S_k$ with $i < j$ we have $\nu_i \leq \nu_j$. Let in the following be $i_k' = \min\{\lfloor n_k/\eta \rfloor + 1, \lfloor \tilde{n}_k/\eta \rfloor + \lfloor n_k/\eta \rfloor + 1\}$. Assume that $B \geq z_{\mathrm{eC-SH}}$, then we have for each round $k$

$$
\begin{aligned}
r_k &\geq \frac{B}{(n_k - \tilde{n}_k)\lceil \log_\eta(n) \rceil} - 1 \\
&\geq \frac{\eta}{n_k - \tilde{n}_k} \max_{i=2,\dots,n} i \left( 1 + \min\left\{ R, \gamma^{-1}\left( \max\left\{ \frac{\epsilon}{4}, \frac{\nu_i - \nu_1}{2} \right\} \right) \right\} \right) - 1 \\
&\geq \frac{\eta}{n_k - \tilde{n}_k} i_k' \left( 1 + \min\left\{ R, \gamma^{-1}\left( \max\left\{ \frac{\epsilon}{4}, \frac{\nu_{i_k'} - \nu_1}{2} \right\} \right) \right\} \right) - 1 \\
&\overset{(*)}{\geq} \frac{\eta}{n_k - \tilde{n}_k} (n_k - \tilde{n}_k)/\eta \left( 1 + \min\left\{ R, \gamma^{-1}\left( \max\left\{ \frac{\epsilon}{4}, \frac{\nu_{i_k'} - \nu_1}{2} \right\} \right) \right\} \right) - 1 \\
&= \left( 1 + \min\left\{ R, \gamma^{-1}\left( \max\left\{ \frac{\epsilon}{4}, \frac{\nu_{i_k'} - \nu_1}{2} \right\} \right) \right\} \right) - 1 \\
&= \min\left\{ R, \gamma^{-1}\left( \max\left\{ \frac{\epsilon}{4}, \frac{\nu_{i_k'} - \nu_1}{2} \right\} \right) \right\},
\end{aligned}
$$

where the fourth line $(*)$ follows from:

- **Case 1:** $i_k' = \lfloor \tilde{n}_k/\eta \rfloor + 1$.
  We have
  $$ i_k' \geq n_k/\eta \geq (n_k - \tilde{n}_k)/\eta. $$

- **Case 2:** $i_k' = \lfloor \tilde{n}_k/\eta \rfloor + \lfloor n_{k-1}/\eta \rfloor + 1$.
  If $\tilde{n}_k = 0$, we have
  $$ i_k' = \left\lfloor \frac{n_{k-1}}{\eta} \right\rfloor + 1 \geq \frac{n_{k-1}}{\eta} \geq \frac{n_k}{\eta} \geq \frac{n_k - \tilde{n}_k}{\eta}. $$

If $\tilde{n}_k \geq 1$, we have

$$
\begin{aligned}
i'_k &\geq \frac{\tilde{n}_k}{\eta} - 1 + \frac{n_{k-1}}{\eta} - 1 + 1 \\
&= \frac{\tilde{n}_k - n_{k-1} - \eta}{\eta} \\
&\geq \frac{n_k + (\eta - 1)n_k + \tilde{n}_k - \eta}{\eta} \\
&\geq \frac{n_k}{\eta} \geq \frac{n_k - \tilde{n}_k}{\eta},
\end{aligned}
$$

where line 3 follows from $n_k = \lfloor n_{k-1}/\eta \rfloor$ and line 4 from $n_k \geq \tilde{n}_k \geq 1$ and $\eta \geq 2$, so we have $\eta - 1 \geq 1$, so we can estimate $n_k \geq 1$.

Next, we show that $\ell_{i,t} - \ell_{1,t} > 0$ for all $t \geq \tau_i := \gamma^{-1}\left(\frac{\nu_i - \nu_1}{2}\right)$. Given the definition of $\gamma$, we have for all $i \in [n]$ that $|\ell_{i,t} - \nu_i| \leq \gamma(t) \leq \frac{\nu_i - \nu_1}{2}$ where the last inequality holds for $t \geq \tau_i$. Thus, for $t \geq \tau_i$ we have

$$
\begin{aligned}
\ell_{i,t} - \ell_{1,t} &= \ell_{i,t} - \nu_i + \nu_i - \nu_1 + \nu_1 - \ell_{1,t} \\
&= \ell_{i,t} - \nu_i - (\ell_{1,t} - \nu_1) + \nu_i - \nu_1 \\
&\geq -2\gamma(t) + \nu_i - \nu_1 \\
&\geq -2\frac{\nu_i - \nu_1}{2} + \nu_i - \nu_1 \\
&= 0.
\end{aligned}
$$

Under this scenario, we will eliminate arm $i$ before arm 1 since on each round the arms are sorted by their empirical losses and the top half are discarded. Note that by the assumption the $\nu_i$ limits are non-decreasing in $i$ so that the $\tau_i$ values are non-increasing in $i$.

Fix a round $k$ and assume $1 \in S_k \cup C_k$ (note, $1 \in S_0 \cup C_0$). The above calculation shows that

$$
t \geq \tau_i \implies \ell_{i,t} \geq \ell_{1,t}. \tag{1}
$$

We regard two different scenarios in the following.

- **Case 1:** $k \leq s - 1$.
  In this case, we keep the best $\lfloor n_k/\eta \rfloor - \lfloor \tilde{n}_k/\eta \rfloor$ arms from the set $S_k \cup C_k \backslash C_{k+1}$ and have already promoted the best $\lfloor \tilde{n}_k/\eta \rfloor$ from $C_k$.

$$
\{1 \in S_k \cup C_k, \ 1 \notin S_{k+1} \cup C_{k+1}\}
$$

$$
\Longleftrightarrow \left\{ \sum_{i \in S_k \cup C_k \backslash C_{k+1}} \mathbf{1}\{\ell_{i,r_k} < \ell_{1,r_k}\} \geq \lfloor n_k/\eta \rfloor - \lfloor \tilde{n}_k/\eta \rfloor, \right.
$$

$$
\left. \sum_{i \in C_k} \mathbf{1}\{\ell_{i,r_k} < \ell_{1,r_k}\} \geq \lfloor \tilde{n}_k/\eta \rfloor \right\}
$$

$$
\Longrightarrow \left\{ \sum_{i \in S_k \cup C_k \backslash C_{k+1}} \mathbf{1}\{r_k < \tau_i\} \geq \lfloor n_k/\eta \rfloor - \lfloor \tilde{n}_k/\eta \rfloor, \right.
$$

$$
\left. \sum_{i \in C_k} \mathbf{1}\{r_k < \tau_i\} \geq \lfloor \tilde{n}_k/\eta \rfloor \right\}
$$

$$
\Longrightarrow \left\{ \sum_{i=2}^{\lfloor n_k/\eta \rfloor - \lfloor \tilde{n}_k/\eta \rfloor + \lfloor \tilde{n}_k/\eta \rfloor + 1} \mathbf{1}\{r_k < \tau_i \wedge i \in S_k \cup C_k \backslash C_{k+1}\} \geq \lfloor n_k/\eta \rfloor \right.
$$

$$
\left. - \lfloor \tilde{n}_k/\eta \rfloor, \quad \sum_{i=2}^{\lfloor \tilde{n}_k/\eta \rfloor + \lfloor n_{k-1}/\eta \rfloor + 1} \mathbf{1}\{r_k < \tau_i \wedge i \in C_k\} \geq \lfloor \tilde{n}_k/\eta \rfloor \right\}
$$

$$\implies \left\{ r_k < \min\left\{ \tau_{\lfloor n_k/\eta \rfloor +1}, \tau_{\lfloor \tilde{n}_k/\eta \rfloor + \lfloor n_{k-1}/\eta \rfloor +1} \right\} \right\}$$

$$\iff \left\{ r_k < \tau_{\max\{\lfloor n_k/\eta \rfloor +1, \lfloor \tilde{n}_k/\eta \rfloor + \lfloor n_{k-1}/\eta \rfloor +1\}} \right\},$$

where the first line follows by the definition of the algorithm and the second by Equation 1. In the third line we assume the worst case scenario, where the best $\lfloor n_k/\eta \rfloor - \lfloor \tilde{n}_k/\eta \rfloor$ arms in $S_k \cup C_k \backslash C_{k+1}$ are all worse than the best $\lfloor \tilde{n}_k/\eta \rfloor$ arms in $C_k$ (which are kept in the set $C_{k+1}$) and vice versa that the best $\lfloor \tilde{n}_k/\eta \rfloor$ arms in $C_k$ are worse than all arms in $S_k$. The fourth line follows by $\tau_i$ being non-increasing (for all $i < j$ we have $\tau_i \geq \tau_j$ and consequently, $\mathbf{1}\{r_k < \tau_i\} \geq \mathbf{1}\{r_k < \tau_j\}$ so the *first* indicators of the sum not including 1 would be on before any other $i$'s in $S_k \subset [n]$ sprinkled throughout $[n]$).

- **Case 2:** $k = s$.
  In this case we keep the best $\lfloor n_k/\eta \rfloor$ arms from $S_k \cup C_k$ and have $C_{k+1} = \emptyset$, thus we get analogously as above

$$\{1 \in S_k \cup C_k, \ 1 \notin S_{k+1}\} \iff \left\{ \sum_{i \in S_k \cup C_k} \mathbf{1}\{\ell_{i,r_k} < \ell_{1,r_k}\} \geq \lfloor n_k/\eta \rfloor \right\}$$

$$\implies \left\{ \sum_{i \in S_k \cup C_k} \mathbf{1}\{r_k < \tau_i\} \geq \lfloor n_k/\eta \rfloor \right\}$$

$$\implies \left\{ \sum_{i=2}^{\lfloor n_k/\eta \rfloor +1} \mathbf{1}\{r_k < \tau_i\} \geq \lfloor n_k/\eta \rfloor \right\}$$

$$\iff \left\{ r_k < \tau_{\lfloor n_k/\eta \rfloor +1} \right\}.$$

Overall, we can conclude, that $1 \in S_k \cup C_k$ and $1 \notin S_{k+1} \cup C_{k+1}$ if $r_k < \tau_{\max\{\lfloor n_k/\eta \rfloor +1, \lfloor \tilde{n}_k/\eta \rfloor + \lfloor n_k/\eta \rfloor +1\}}$. This implies

$$\{1 \in S_k \cup C_k, \ r_k \geq \tau_{i_k'}\} \implies \{1 \in S_{k+1} \cup C_{k+1}\}. \tag{2}$$

Recalling that $r_k \geq \gamma^{-1}\big( \max\big\{ \frac{\epsilon}{4}, \frac{\nu_{i_k'} - \nu_1}{2} \big\} \big)$ and $\tau_{i_k'} = \gamma^{-1}\big( \frac{\nu_{i_k'} - \nu_1}{2} \big)$, we examine the following three exhaustive cases:

- **Case 1:** $\frac{\nu_{i_k'} - \nu_1}{2} \geq \frac{\epsilon}{4}$ and $1 \in S_k \cup C_k$

  In this case, $r_k \geq \gamma^{-1}\big( \frac{\nu_{i_k'} - \nu_1}{2} \big) = \tau_{i_k'}$. By Equation 2 we have that $1 \in S_{k+1} \cup C_{k+1}$ since $1 \in S_k \cup C_k$.

- **Case 2:** $\frac{\nu_{i_k'} - \nu_1}{2} < \frac{\epsilon}{4}$ and $1 \in S_k \cup C_k$

  In this case $r_k \geq \gamma^{-1}\big( \frac{\epsilon}{4} \big)$ but $\gamma^{-1}\big( \frac{\epsilon}{4} \big) < \tau_{i_k'}$. Equation 2 suggests that it may be possible for $1 \in S_k \cup C_k$ but $1 \notin S_{k+1} \cup C_{k+1}$. On the good event that $1 \in S_{k+1} \cup C_{k+1}$, the algorithm continues and on the next round either case 1 or case 2 could be true. So assume $1 \notin S_{k+1} \cup C_{k+1}$. Here we show that $\{1 \in S_k \cup C_k, \ 1 \notin S_{k+1} \cup C_{k+1}\} \implies \max_{i \in S_{k+1} \cup C_{k+1}} \nu_i \leq \nu_1 + \epsilon/2$. Because $1 \in S_0 \cup C_0$, this guarantees that Algorithm 1 either exits with arm $\widehat{i} = 1$ or some arm $\widehat{i}$ satisfying $\nu_{\widehat{i}} \leq \nu_1 + \epsilon/2$.

  Let $p = \min\{i \in [n] : \frac{\nu_i - \nu_1}{2} \geq \frac{\epsilon}{4}\}$. Note that $p > i_k'$ by the criterion of the case and

$$r_k \geq \gamma^{-1}\left( \frac{\epsilon}{4} \right) \geq \gamma^{-1}\left( \frac{\nu_i - \nu_1}{2} \right) = \tau_i, \quad \forall i \geq p.$$

  Thus, by Equation 1 ($t \geq \tau_i \implies \ell_{i,t} \geq \ell_{1,t}$) we have that arms $i \geq p$ would always have $\ell_{i,r_k} \geq \ell_{1,r_k}$ and be eliminated before or at the same time as arm 1, presuming $1 \in S_k \cup C_k$. In conclusion, if arm 1 is eliminated so that $1 \in S_k \cup C_k$ but $1 \notin S_{k+1} \cup C_{k+1}$ then $\max_{i \in S_{k+1} \cup C_{k+1}} \nu_i \leq \max_{i < p} \nu_i < \nu_1 + \epsilon/2$ by the definition of $p$.

- **Case 3:** $1 \notin S_k \cup C_k$

  Since $1 \in S_0 \cup C_0$, there exists some $r < k$ such that $1 \in S_r \cup C_r$ and $1 \notin S_{r+1} \cup C_{r+1}$. For this $r$, only case 2 is possible since case 1 would proliferate $1 \in S_{r+1} \cup C_{r+1}$. However, under case 2, if $1 \notin S_{r+1} \cup C_{r+1}$ then $\max_{i \in S_{r+1} \cup C_{r+1}} \nu_i \leq \nu_1 + \epsilon/2$.

Because $1 \in S_0 \cup C_0$, we either have that 1 remains in $S_k \cup C_k$ (possibly alternating between cases 1 and 2) for all $k$ until the algorithm exits with the best arm 1, or there exists some $k$ such that case 3 is true and the algorithm exits with an arm $\widehat{i}$ such that $\nu_{\widehat{i}} \leq \nu_1 + \epsilon/2$.

**Part II: iSHA variants analysis**

Next, we proof the same guarantee for the discarding and preserving Incremental-SuccessiveHalving algorithms given in Algorithm 3 and Algorithm 4.
Therefore we proceed in two steps: First, we will reduce the `d-iSHA` algorithm to the SH algorithm to take over its theoretical guarantees. Second, we will show where the proof of SH has to be modified to achieve the same theoretical guarantees for our `p-iSHA` algorithm.

Step 1: We will distinguish two different cases in the following in order to reduce the discarding Incremental-SuccessiveHalving algorithm 3 to the original version of Successive Halving by Jamieson & Talwalkar (2016a) (or Karnin et al. (2013)).

- **Case 1:** $(C_k)_k = \emptyset$.
  If we have $(C_k)_k = \emptyset$, we have simply the Successive Halving algorithm by Jamieson & Talwalkar (2016a) and can keep their theoretical guarantees.

- **Case 2:** $(C_k)_k \neq \emptyset$.
  Thus the interesting case which we will consider in the following is the case $(C_k)_k \neq \emptyset$. Assume that Algorithm 3 is called as subroutine by Algorithm 2. Since $(C_k)_k \neq \emptyset$, Algorithm 3 was already called before with number of arms $\tilde{n}$ and budget $\tilde{r}_s = \tilde{R}/\eta^{\tilde{s}} = \frac{R}{\eta}/\eta^{s-1} = R/\eta^s = r_k$ for $s \in \{0, \ldots, \lfloor \log_\eta(R) \rfloor\}$. Thus, the arms in $(C_k)_k$ were already pulled for $r_k$ times and their loss values $(L_k)_k$ were observed. Combining these with the loss values we observe in each iteration $k$ in Algorithm 3 for $r_k$ pulls of the arms in $S_k \backslash C_k$, we can keep the best $\lfloor n/\eta^{k+1} \rfloor$ arms from $S_k$ regarding the observed losses of the recent pulls of $S_k \backslash C_k$ and the before observed losses of $C_k$. Therefore, we get the same arms in $S_{k+1}$ as starting Algorithm 4 from scratch with $(C_k)_k = \emptyset$ and $S = S \cup C_0$ and can apply Case 1.

To conclude both cases, we can keep the theoretical result that was proven by Li et al. (2018) for the original version of Successive Halving in a finite horizon setting ($R < \infty$).

Step 2: To achieve the same guarantee for the preserving Incremental-SuccessiveHalving algorithm, we can proceed analogue as in the proof of Successive Halving by Li et al. (2018). For a fixed round $k$ and $1 \in S_k \cup C_k$, since $1 \in S_0 \cup C_0$, we have

$$\{1 \in S_k \cup C_k, 1 \notin S_{k+1}\} \Leftrightarrow \left\{ \sum_{i \in S_k \cup C_k} \mathbf{1}\{\ell_{i,r_k} < \ell_{1,r_k}\} \geq \lfloor n_k/\eta \rfloor \right\}$$

$$\Rightarrow \left\{ \sum_{i \in S_k \cup C_k} \mathbf{1}\{r_k < \tau_i\} \geq \lfloor n_k/\eta \rfloor \right\}$$

$$\Rightarrow \left\{ \sum_{i=2}^{n_k + |C_k \backslash (S_k \cap C_k)| + 1} \mathbf{1}\{r_k < \tau_i\} \geq \lfloor n_k/\eta \rfloor \right\}$$

$$\Leftrightarrow \{r_k < \tau_{\lfloor n_k/\eta \rfloor + 1}\}.$$

The rest of the proof is the same as that for Successive Halving in Li et al. (2018). □

### B.2 COMPARISON OF SHA AND iSHA

*Proof of Theorem 6.5.* Let us first regard the number of total pulls when we run $\text{SHA}(n, r)$ in comparison to a run of $\text{iSHA}(n, r)$, where we assume that we had already run $\text{SHA}(\tilde{n}, \tilde{r})$. We concentrate in the following on a lower bound on the pulls of $\text{SHA}(n, r)$.

$$
\begin{aligned}
\sum_{k=0}^{s} n_k r_k &= \sum_{k=0}^{s} \left\lfloor \frac{n}{\eta^k} \right\rfloor \cdot \left\lfloor \frac{R\eta^k}{\eta^s} \right\rfloor \\
&\geq \sum_{k=0}^{s} \left( \frac{n}{\eta^k} - 1 \right) \left( \frac{R\eta^k}{\eta^s} - 1 \right) \\
&= \sum_{k=0}^{s} \frac{nR}{\eta^s} - \frac{R\eta^k}{\eta^s} - \frac{n}{\eta^k} + 1 \\
&= \frac{(s+1)(nR+\eta^s)}{\eta^s} - \frac{R}{\eta^s} \sum_{k=0}^{s} \eta^k - n \sum_{k=0}^{s} \left( \frac{1}{\eta} \right)^k \\
&= \frac{(s+1)(nR+\eta^s)}{\eta^s} - \frac{R(\eta^{s+1}-1)}{\eta^s(\eta-1)} - \underbrace{\frac{n(1-(1/\eta)^{s+1})}{1-1/\eta}}_{=\frac{n\left(\frac{\eta^{s+1}-1}{\eta^{s+1}}\right)}{\frac{\eta-1}{\eta}} = \frac{n(\eta^{s+1}-1)}{\eta^s(\eta-1)}} \\
&= \frac{(s+1)(nR+\eta^s)}{\eta^s} - \frac{(\eta^{s+1}-1)(R+n)}{\eta^s(\eta-1)} \\
&= \frac{(s+1)(nR+\eta^s)(\eta-1) - (\eta^{s+1}-1)(R+n)}{\eta^s(\eta-1)},
\end{aligned}
$$

where we used the closed form for the geometric series in the fifth line and simple transformations in all other lines.

An upper bound on the total pulls of $\text{iSHA}(n, r)$ is given by

$$
\begin{aligned}
\sum_{k=0}^{s} n_k r_k &= \sum_{k=0}^{s} \left( \lfloor n/\eta^k \rfloor - \lfloor \tilde{n}/\eta^k \rfloor \right) \left\lfloor \frac{R\eta^k}{\eta^s} \right\rfloor \\
&\leq \sum_{k=0}^{s} \left( \frac{n-\tilde{n}}{\eta^k} + 1 \right) \cdot \frac{R\eta^k}{\eta^s} \\
&= \frac{(s+1)(n-\tilde{n})R}{\eta^s} + \frac{R}{\eta^s} \sum_{k=0}^{s} \eta^k \\
&= \frac{(s+1)(n-\tilde{n})R}{\eta^s} + \frac{R(\eta^{s+1}-1)}{\eta^s(\eta-1)} \\
&= \frac{(s+1)(n-\tilde{n})R(\eta-1) + R(\eta^{s+1}-1)}{\eta^s(\eta-1)}.
\end{aligned}
$$

Finally, we compare both by building the quotient

$$
\begin{aligned}
\frac{\#\{\text{total pulls of iSHA}\,(n,r)\}}{\#\{\text{total pulls of SH}(n,r)\}} &\leq \frac{(s+1)(n-\tilde{n})R(\eta-1) + R(\eta^{s+1}-1)}{(s+1)(nR+\eta^s)(\eta-1) - (\eta^{s+1}-1)(R+n)} \\
&= 1 - \frac{(s+1)(\tilde{n}R+\eta^s)(\eta-1) - (\eta^{s+1}-1)(2R+n)}{(s+1)(nR+\eta^s)(\eta-1) - (\eta^{s+1}-1)(R+n)}.
\end{aligned}
$$

$\square$

It is worth mentioning that we can do a similar analysis for the discarding and preserving Incremental-SuccessiveHalving algorithms given in Algorithm 3 and Algorithm 4:

Analogously as in the proof of Theorem 6.5, we first need an upper bound on the total pulls in a run of d-iSHA$(n, r)$. While we only sample $n - \tilde{n}$ new arms in the first round of d-iSHA, the best $n/\eta$ arms may be all from the newly sampled ones and thus none of the arms which are kept into the next round of d-iSHA was already pulled with a higher budget in the run of SH$(\tilde{n}, \tilde{r})$. In this worst case, we can estimate

$$
\sum_{k=0}^{s} n_k r_k = (n - \tilde{n}) \left\lfloor \frac{R}{\eta^s} \right\rfloor + \sum_{k=1}^{s} \left\lfloor \frac{n}{\eta^k} \right\rfloor \left\lfloor \frac{R\eta^k}{\eta^s} \right\rfloor
$$

$$
\leq \frac{(n - \tilde{n})R}{\eta^s} + \sum_{k=1}^{s} \frac{nR}{\eta^s}
$$

$$
= \frac{(n - \tilde{n})R}{\eta^s} + \frac{snR}{\eta^s}
$$

$$
= \frac{R((s+1)n - \tilde{n})}{\eta^s}.
$$

Again, we can now compute the quotient of the pulls as follows.

$$
\frac{\#\{\text{total pulls of d-iSHA } (n, r)\}}{\#\{\text{total pulls of SH}(n, r)\}} \leq \frac{(\eta - 1)R((s+1)n - \tilde{n})}{(\eta - 1)(s+1)(nR + \eta^s) - (\eta^{s+1} - 1)(R + n)}
$$

$$
= 1 - \frac{(\eta - 1)((s+1)\eta^s + R\tilde{n}) - (\eta^{s+1} - 1)(R + n)}{(\eta - 1)(s+1)(nR + \eta^s) - (\eta^{s+1} - 1)(R + n)}.
$$

Note that we can apply the same for the number of pulls of p-iSHA since we have the same worst-case scenario where we only keep newly sampled configurations into the next round of p-iSHA and none of the previously promoted configurations.

To get an intuition for the improvement in the number of total pulls, we show in Figure 5 and Figure 6 the above terms for different values of rounds $s$, maximal budgets per round $R$ and discarding portion $\eta$. Note that the above results assume the worst-case scenario for the p-iSHA resp. the d-iSHA algorithm in which all previously promoted configurations perform worse than all newly sampled ones. In most problem scenarios the average improvement in the number of total pulls of p-iSHA resp. d-iSHA will lie between the plotted curves of the worst case scenario in Figure 5 and the best case scenario which coincidences with iSHA and is shown in Figure 6. Since our proposed methods will never need a greater number of total pulls than SH, we plotted the minimum value of 1 and our derived fractions in Theorem 6.5.

*Proof of Corollary 6.6.* In the following, we only regard the asymptotic behavior of the number of pulls for an infinite large budget. In this case, we can ignore the flooring functions since the asymptotic behavior is not affected by those. We get for the asymptotic ratio between the number of pulls of rerunning SH and iSHA that

$$
\lim_{s \to \infty} \frac{\sum_{k=0}^{s} \left( \lfloor n/\eta^k \rfloor - \lfloor \tilde{n}/\eta^k \rfloor \right) \left\lfloor \frac{R\eta^k}{\eta^s} \right\rfloor}{\sum_{k=0}^{s} \left\lfloor \frac{n}{\eta^k} \right\rfloor \cdot \left\lfloor \frac{R\eta^k}{\eta^s} \right\rfloor} = \lim_{s \to \infty} \frac{\sum_{k=0}^{s} \frac{(n - \tilde{n})R}{\eta^s}}{\sum_{k=0}^{s} \frac{nR}{\eta^s}} = \frac{n - \tilde{n}}{n}
$$

$$
= 1 - \eta^{-1},
$$

where we used that in each new run of iSHA it holds that $n = |S| + |C_0| = |C_0| \cdot \eta$, where $C_0$ is the number of configurations in the previous run with cardinality $\tilde{n}$. $\square$

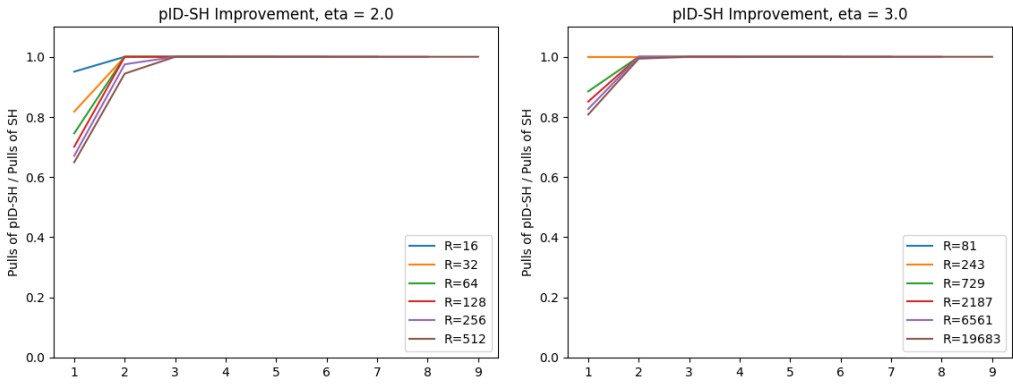

Figure 5: Fraction of the number of total pulls of `p-iSHA` resp. `d-iSHA` and SHA for different values of rounds of SH $s$, maximal budgets per round $R$ and discarding fraction $\eta$.

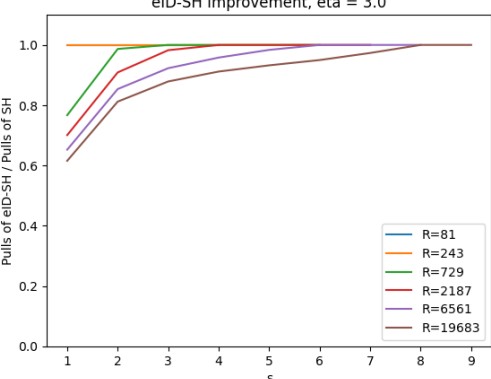

Figure 6: Fraction of number of total pulls of `iSHA` for different values of rounds of SHA $s$ and maximal budgets per round $R$.

### B.3 INCREMENTAL-HYPERBAND

*Proof of Theorem 6.8.* To derive the necessary budget of iHB in Algorithm 2, we simply have to sum up all necessary budgets for each call of xiSHA. Luckily, the necessary budgets for iSHA, d-iSHA and p-iSHA do not differ, thus a run of iHB uses independent of the variant of the called Successive Halving algorithm a total budget of

$$\sum_{s=0}^{\lfloor \log_\eta(R) \rfloor} \text{Budget\_xiSHA}(n_s, r_s)$$

$$= \sum_{s=0}^{\lfloor \log_\eta(R) \rfloor} \eta \left\lceil \log_\eta \left( \left\lceil (\lfloor \log_\eta(R) \rfloor + 1) \cdot \frac{\eta^s}{s+1} \right\rceil \right) \right\rceil$$

$$\cdot \max_{i=2,\dots,n_s} i \left( 1 + \min \left\{ R, \gamma^{-1} \left( \max \left\{ \frac{\epsilon}{4}, \frac{\nu_i - \nu_1}{2} \right\} \right) \right\} \right)$$

$$= (*).$$

Due to simple estimates and transformations, we get

$$\eta \left\lceil \log_\eta \left( \left\lceil (\lfloor \log_\eta(R) \rfloor + 1) \cdot \frac{\eta^s}{s+1} \right\rceil \right) \right\rceil \leq \eta \left\lceil \log_\eta \left( \left\lceil (\log_\eta(R) + 1) \right\rceil \cdot \left\lceil \frac{\eta^s}{s+1} \right\rceil \right) \right\rceil$$

$$= \eta \left\lceil \log_\eta \left( \left\lceil \log_\eta(R) + 1 \right\rceil \right) + \log_\eta \left( \left\lceil \frac{\eta^s}{s+1} \right\rceil \right) \right\rceil$$

$$\leq \eta \left\lceil \log_\eta \left( \log_\eta(R) + 2 \right) + \log_\eta \left( \frac{\eta^s}{s+1} + 1 \right) \right\rceil$$

$$\leq \eta \left\lceil \log_\eta \left( \log_\eta(R) \right) + 2 + \log_\eta \left( \frac{\eta^s}{s+1} \right) + 1 \right\rceil$$

$$= \eta \left\lceil \log_\eta \left( \log_\eta(R) \right) + \log_\eta \left( \eta^s \right) - \log_\eta \left( s+1 \right) + 3 \right\rceil$$

$$\leq \eta \left( \log_\eta \left( \log_\eta(R) \right) + s - \log_\eta \left( s+1 \right) + 4 \right).$$

Note that the fourth line follows from

$$\log_\eta(x+1) \leq \log_\eta(x) + 1$$
$$\Leftrightarrow x + 1 \leq \eta \cdot x$$
$$\Leftrightarrow x \geq \frac{1}{\eta - 1}.$$

In our setting, we have $\eta \geq 2$, thus $\log_\eta(x+1) \geq \log_\eta(x) + 1$ if and only if $x \geq 1$. We have $\frac{\eta^s}{s+1} \geq 1$ for $s \geq 0$ and also wlog. $\log_\eta(R) \geq 2$, otherwise the value of $s_{max}$ and thus the run of Hyperband would be trivial.
We can continue with

$$(*) \leq \sum_{s=0}^{\lfloor \log_\eta(R) \rfloor} \eta \left( \log_\eta \left( \log_\eta(R) \right) + s - \log_\eta \left( s+1 \right) + 4 \right)$$

$$\cdot \underbrace{\max_{s=0,\dots,\lfloor \log_\eta(R) \rfloor} \max_{i=2,\dots,n_s} i \left( 1 + \min \left\{ R, \gamma^{-1} \left( \max \left\{ \frac{\epsilon}{4}, \frac{\nu_i - \nu_1}{2} \right\} \right) \right\} \right)}_{=: \bar{\gamma}^{-1}}$$

$$= \eta \left( (\lfloor \log_\eta(R) \rfloor + 1) \left( \log_\eta(\log_\eta(R)) + 4 \right) + \sum_{s=0}^{\lfloor \log_\eta(R) \rfloor} s - \sum_{s=0}^{\lfloor \log_\eta(R) \rfloor} \log_\eta(s+1) \right) \bar{\gamma}^{-1}$$

$$= \eta \Bigg( (\lfloor \log_\eta(R) \rfloor + 1) \left( \log_\eta(\log_\eta(R)) + 4 \right) + \frac{\lfloor \log_\eta(R) \rfloor \left( \lfloor \log_\eta(R) \rfloor + 1 \right)}{2}$$

$$- \log_\eta \left( \prod_{s=0}^{\lfloor \log_\eta(R) \rfloor} (s+1) \right) \Bigg) \bar{\gamma}^{-1}$$

$$= \eta \Bigg( \left( \lfloor \log_\eta(R) \rfloor + 1 \right) \left( \log_\eta(\log_\eta(R)) + 4 \right) + \frac{\lfloor \log_\eta(R) \rfloor \left( \lfloor \log_\eta(R) \rfloor + 1 \right)}{2}$$

$$- \log_\eta \left( \left( \lfloor \log_\eta(R) \rfloor + 1 \right)! \right) \Bigg) \bar{\gamma}^{-1}.$$

Since we choose the budget $B$ in our iHB algorithm as $B = (s_{\max} + 1)R = \left( \lfloor \log_\eta(R) \rfloor + 1 \right) R$, we can divide both by $\left( \lfloor \log_\eta(R) \rfloor + 1 \right)$ and get

$$R \geq \eta \left( \log_\eta(\log_\eta(R)) + 4 + \frac{\lfloor \log_\eta(R) \rfloor}{2} - \frac{\log_\eta \left( \left( \lfloor \log_\eta(R) \rfloor + 1 \right)! \right)}{\lfloor \log_\eta(R) \rfloor + 1} \right) \bar{\gamma}^{-1}.$$

Recall that in each call of xiSHA in round $s$ of iHB we compare $n_s = \left( \lfloor \log_\eta(R) \rfloor + 1 \right) \frac{\eta^s}{s+1}$ hyperparameter configurations, thus we get an overall number of samples of

$$\left( \lfloor \log_\eta(R) \rfloor + 1 \right) \sum_{s=0}^{\lfloor \log_\eta(R) \rfloor} \frac{\eta^s}{s+1}$$

$$\geq \sum_{s=0}^{\lfloor \log_\eta(R) \rfloor} \eta^s$$

$$\overset{\text{Geometric Sum}}{=} \frac{\eta^{\lfloor \log_\eta(R) \rfloor + 1} - 1}{\eta - 1}$$

$$\geq \frac{R - 1}{\eta - 1}.$$

By assumption 6.7 we have an $\epsilon$-optimal hyperparameter configuration in our sample set with probability at least $1 - \delta$ if and only if

$$\frac{R - 1}{\eta - 1} \geq \left\lceil \log_{1-\alpha}(\delta) \right\rceil$$

$$\Leftrightarrow \quad R \geq \left\lceil \log_{1-\alpha}(\delta) \right\rceil (\eta - 1) + 1.$$

$\square$

## C  THEORETICAL ANALYSIS OF ASHA

*Proof of Theorem 6.2.* For sake of simplicity, we denote the configurations by their indices and assume without loss of generality that the configurations are ordered, such that the optimal configuration has index 1, the second best 2 etc. In the worst case, we observe the configurations in such an order that we have never two consistent rankings in two successive rungs, thus we have to enlarge the rung each time we have at least $\eta$ configurations in the recent top rung $K$. With this, we get in each rung $k$ at least $\lfloor n/\eta^k \rfloor$ configurations for which the budget $b_k = r\eta^k$ is used by algorithm design. In addition, we can bound the index of the top rung $K$ by $\lfloor \log_\eta(n) \rfloor$, because we have at least $\eta$ times many configurations in rung $k+1$ in comparison to rung $k$ and the first rung has index 0. We can compute the following upper bound for the budget of ASHA:

$$
\begin{aligned}
B &= \sum_{k=0}^{K} |\text{rung}_k| \cdot b_k \\
&\leq \sum_{k=0}^{K} \left\lfloor \frac{n}{\eta^k} \right\rfloor \cdot r\eta^k \\
&\leq \sum_{k=0}^{K} r \cdot n \\
&= (K+1) \cdot rn \\
&\leq (\lfloor \log_\eta(n) \rfloor + 1)rn.
\end{aligned}
$$

By simple transformations we get

$$
r \geq \frac{B}{(\lfloor \log_\eta(n) \rfloor + 1)n}.
$$

We prove the correctness of ASHA indirectly, so we assume in the following that ASHA does not return the near-optimal configuration. For sake of convenience, we write $[K-1]_0 = \{0, 1, 2, \ldots, K-1\}$. We can regard two different cases for this scenario.

- **Case 1:** Configuration 1 does not even reach the top rung $K$.

$$
1 \notin \text{rung}_K \iff \exists k \in [K-1]_0 : 1 \in \text{rung}_k \land 1 \notin \text{rung}_{k+1}
$$

$$
\iff \exists k \in [K-1]_0 : \sum_{i \in \text{rung}_k \setminus \{1\}} \mathbf{1}\{\ell_{1,b_k} > \ell_{i,b_k}\} > \frac{|\text{rung}_k|}{\eta}
$$

$$
\iff \exists k \in [K-1]_0 : \sum_{i \in \text{rung}_k \setminus \{1\}} \mathbf{1}\{\nu_i - \nu_1 < \nu_i - \ell_{i,b_k} - \nu_1 + \ell_{1,b_k}\} > \frac{|\text{rung}_k|}{\eta}
$$

$$
\Rightarrow \exists k \in [K-1]_0 : \sum_{i \in \text{rung}_k \setminus \{1\}} \mathbf{1}\{\nu_i - \nu_1 < |\nu_i - \ell_{i,b_k}| + |\nu_1 - \ell_{1,b_k}|\} > \frac{|\text{rung}_k|}{\eta}
$$

$$
\Rightarrow \exists k \in [K-1]_0 : \sum_{i \in \text{rung}_k \setminus \{1\}} \mathbf{1}\{\nu_i - \nu_1 < 2\gamma(b_k)\} > \frac{|\text{rung}_k|}{\eta}
$$

$$
\Rightarrow \exists k \in [K-1]_0 : \nu_{\lfloor |\text{rung}_k|/\eta \rfloor + 1} - \nu_1 < 2\gamma(b_k)
$$

$$
\Rightarrow \exists k \in [K-1]_0 : r\eta^k = b_k < \gamma^{-1}\left(\frac{\nu_{\lfloor |\text{rung}_k|/\eta \rfloor + 1} - \nu_1}{2}\right)
$$

$$
\iff \exists k \in [K-1]_0 : r < \eta^{-k}\gamma^{-1}\left(\frac{\nu_{\lfloor |\text{rung}_k|/\eta \rfloor + 1} - \nu_1}{2}\right)
$$

$$
\Rightarrow \frac{B}{(\lfloor \log_\eta(n) \rfloor + 1)n} \leq r < \max_{k \in [K]} \eta^{-k}\gamma^{-1}\left(\frac{\nu_{\lfloor |\text{rung}_{k-1}|/\eta \rfloor + 1} - \nu_1}{2}\right)
$$

$$
\Rightarrow B < (\lfloor \log_\eta(n) \rfloor + 1)n \max_{k \in [K]} \eta^{-k}\gamma^{-1}\left(\frac{\nu_{\lfloor |\text{rung}_{k-1}|/\eta \rfloor + 1} - \nu_1}{2}\right).
$$

By contradiction, we get the following condition for the budget of ASHA to ensure that configuration 1 gets promoted into the top rung $K$:

$$B \geq (\lfloor \log_\eta(n) \rfloor + 1)n \max_{k \in [K]} \eta^{-k} \gamma^{-1} \left( \frac{\nu_{\lfloor |\text{rung}_{k-1}|/\eta \rfloor + 1} - \nu_1}{2} \right).$$

- **Case 2:** Configuration 1 is contained in the top rung $K$, but is not returned by ASHA.

$$\text{return(ASHA)} \neq \{1\} \wedge 1 \in \text{rung}_K$$
$$\Leftrightarrow \exists i \in \text{rung}_K \backslash \{1\} \ : \ \ell_{1,b_K} > \ell_{i,b_K}$$
$$\Leftrightarrow \exists i \in \text{rung}_K \backslash \{1\} \ : \ \nu_i - \nu_1 < \nu_i - \ell_{i,b_K} - \nu_1 + \ell_{1,b_K}$$
$$\Rightarrow \exists i \in \text{rung}_K \backslash \{1\} \ : \ \nu_i - \nu_1 < |\nu_i - \ell_{i,b_K}| + |\nu_1 - \ell_{1,b_K}|$$
$$\Rightarrow \exists i \in \text{rung}_K \backslash \{1\} \ : \ \nu_i - \nu_1 < 2\gamma(b_K)$$
$$\Rightarrow \exists i \in \text{rung}_K \backslash \{1\} \ : \ r\eta^K = b_K < \gamma^{-1} \left( \frac{\nu_i - \nu_1}{2} \right)$$
$$\Rightarrow \exists i \in \text{rung}_K \backslash \{1\} \ : \ \frac{B}{(\lfloor \log_\eta(n) \rfloor + 1)n} \leq r < \eta^{-K} \gamma^{-1} \left( \frac{\nu_i - \nu_1}{2} \right)$$
$$\Rightarrow B < (\lfloor \log_\eta(n) \rfloor + 1)n\eta^{-K} \max_{i \in \text{rung}_K \backslash \{1\}} \gamma^{-1} \left( \frac{\nu_i - \nu_1}{2} \right).$$

By contradiction, ASHA returns a near-optimal solution if it is contained in the top rung $K$ and if

$$B \geq (\lfloor \log_\eta(n) \rfloor + 1)n\eta^{-K} \max_{i \in \text{rung}_K \backslash \{1\}} \gamma^{-1} \left( \frac{\nu_i - \nu_1}{2} \right).$$

To summarize both cases, ASHA needs a budget of

$$B \geq (\lfloor \log_\eta(n) \rfloor + 1)n \cdot \max \left\{ \max_{k \in [K]} \eta^{-k} \gamma^{-1} \left( \frac{\nu_{\lfloor |\text{rung}_{k-1}|/\eta \rfloor + 1} - \nu_1}{2} \right), \eta^{-K} \max_{i \in \text{rung}_K \backslash \{1\}} \gamma^{-1} \left( \frac{\nu_i - \nu_1}{2} \right) \right\}$$

to ensure that the optimal configuration will be returned. $\square$

# D    DETAILED EMPIRICAL RESULTS

In this section we provide the results of the empirical study in more detail, providing individual figures for every benchmark set and value for $\eta$.

## D.1    RESULTS FOR LCBENCH

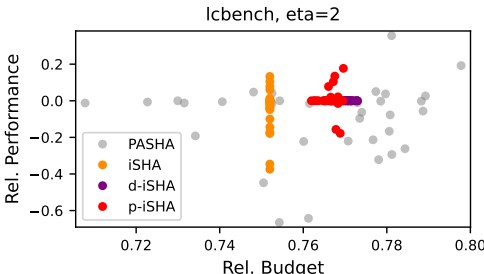

| $\eta = 2$ | Performance | | | Budget | |
|---|---|---|---|---|---|
| Approach | Impr | Degr | Tie | Mean | Std |
| iSHA | 2 | 6 | 26 | 0.752 | 0.0 |
| p-iSHA | 3 | 2 | 29 | 0.7667 | 0.0019 |
| d-iSHA | 0 | 0 | 34 | 0.7695 | 0.0024 |
| PASHA | 2 | 13 | 19 | 0.7642 | 0.0215 |

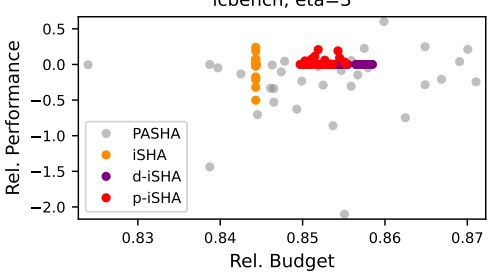

| $\eta = 3$ | Performance | | | Budget | |
|---|---|---|---|---|---|
| Approach | Impr | Degr | Tie | Mean | Std |
| iSHA | 4 | 4 | 26 | 0.8443 | 0.0 |
| p-iSHA | 5 | 0 | 29 | 0.8524 | 0.0015 |
| d-iSHA | 0 | 0 | 34 | 0.8543 | 0.0022 |
| PASHA | 4 | 18 | 12 | 0.8528 | 0.0102 |

## D.2    RESULTS FOR RBV2_XGBOOST BENCHMARK

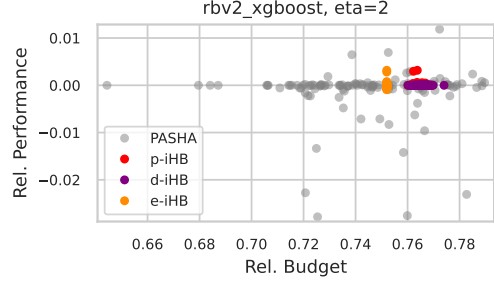

| $\eta = 2$ | Performance | | | Budget | |
|---|---|---|---|---|---|
| Approach | Impr | Degr | Tie | Mean | Std |
| iSHA | 4 | 2 | 113 | 0.752 | 0.0 |
| p-iSHA | 2 | 0 | 117 | 0.7646 | 0.0019 |
| d-iSHA | 0 | 0 | 119 | 0.7654 | 0.0021 |
| PASHA | 11 | 24 | 84 | 0.7451 | 0.0252 |

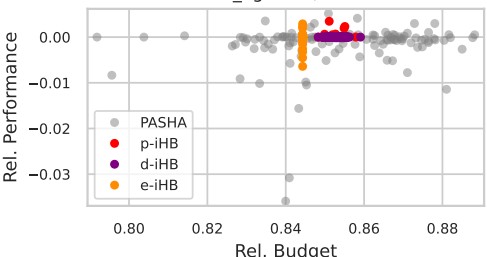

| $\eta = 3$ | Performance | | | Budget | |
|---|---|---|---|---|---|
| Approach | Impr | Degr | Tie | Mean | Std |
| iSHA | 3 | 7 | 109 | 0.8443 | 0.0 |
| p-iSHA | 3 | 0 | 116 | 0.8525 | 0.0018 |
| d-iSHA | 0 | 0 | 119 | 0.8527 | 0.0018 |
| PASHA | 12 | 42 | 65 | 0.8536 | 0.0185 |

## D.3   RESULTS FOR RBV2_RANGER BENCHMARK

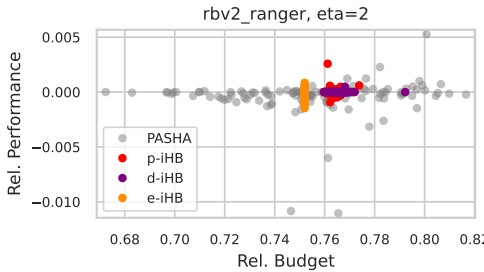

| $\eta = 2$ | Performance | | | Budget | |
|---|---|---|---|---|---|
| Approach | Impr | Degr | Tie | Mean | Std |
| iSHA | 1 | 3 | 115 | 0.752 | 0.0 |
| p-iSHA | 1 | 1 | 117 | 0.7636 | 0.002 |
| d-iSHA | 0 | 0 | 119 | 0.7648 | 0.0036 |
| PASHA | 5 | 18 | 96 | 0.7557 | 0.0294 |

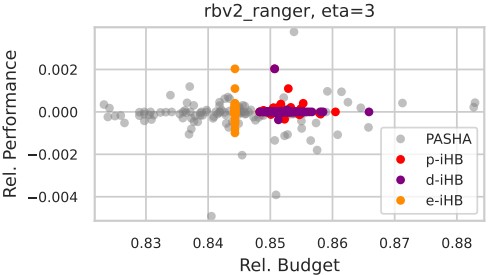

| $\eta = 3$ | Performance | | | Budget | |
|---|---|---|---|---|---|
| Approach | Impr | Degr | Tie | Mean | Std |
| iSHA | 2 | 2 | 115 | 0.8443 | 0.0 |
| p-iSHA | 2 | 0 | 117 | 0.8517 | 0.0019 |
| d-iSHA | 1 | 0 | 118 | 0.8521 | 0.0022 |
| PASHA | 7 | 12 | 100 | 0.8447 | 0.011 |

## D.4   RESULTS FOR RBV2_SVM BENCHMARK

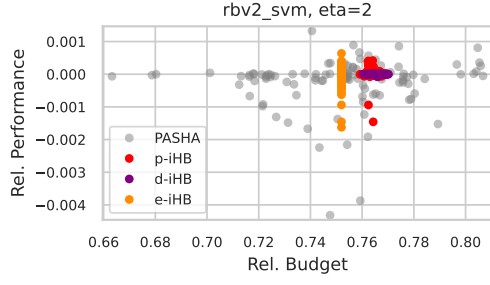

| $\eta = 2$ | Performance | | | Budget | |
|---|---|---|---|---|---|
| Approach | Impr | Degr | Tie | Mean | Std |
| iSHA | 0 | 3 | 103 | 0.752 | 0.0 |
| p-iSHA | 0 | 2 | 104 | 0.7646 | 0.0019 |
| d-iSHA | 0 | 0 | 106 | 0.7654 | 0.002 |
| PASHA | 3 | 14 | 89 | 0.7522 | 0.0273 |

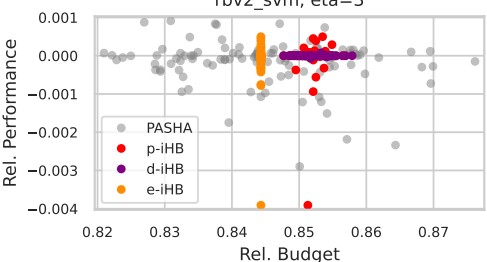

| $\eta = 3$ | Performance | | | Budget | |
|---|---|---|---|---|---|
| Approach | Impr | Degr | Tie | Mean | Std |
| iSHA | 0 | 1 | 105 | 0.8443 | 0.0 |
| p-iSHA | 0 | 2 | 104 | 0.8519 | 0.0016 |
| d-iSHA | 0 | 0 | 106 | 0.8522 | 0.0018 |
| PASHA | 1 | 16 | 89 | 0.8448 | 0.012 |

## D.5 DIRECT COMPARISON OF iSHA TO PASHA

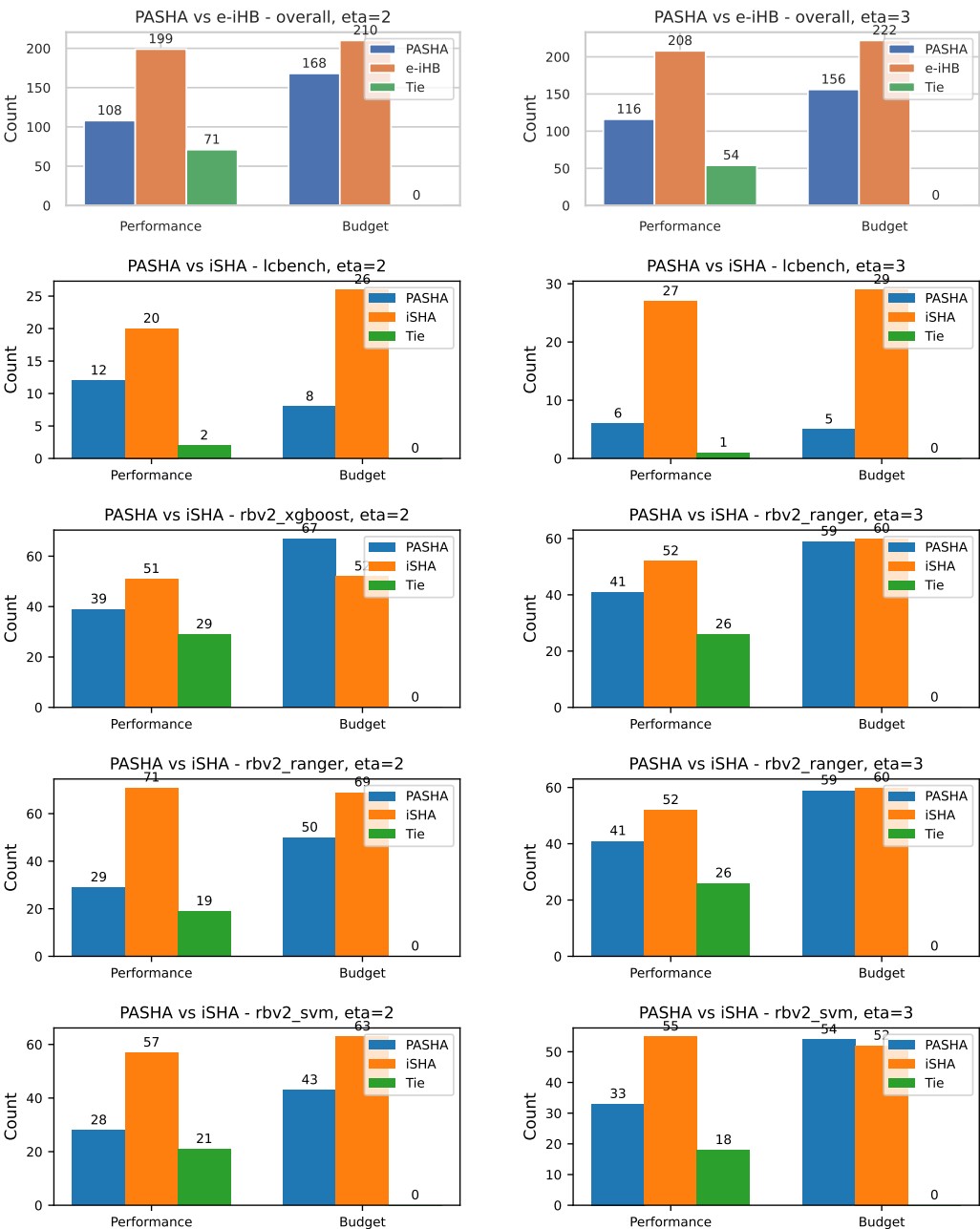

Figure 7: Direct comparison of iSHA to the state of the art PASHA counting for how many of the considered benchmark instances which method yielded a better performance or used less budget. We present bar charts for an overall impression including all evaluated benchmark instances and individually for every benchmark set.