# OpenReview forum: "Incremental Successive Halving for Hyperparameter Optimization with Budget Constraints"
_ICLR.cc/2024/Conference — Submitted to ICLR 2024_

### Official Review · Reviewer_nzDd · 2023-10-26

**Soundness:** 2 fair
**Presentation:** 1 poor
**Contribution:** 2 fair
**Rating:** 3
**Confidence:** 4

**Summary:**

The work introduces a way on how to extend the maximal budget for Successive Halving without starting from scratch, but by reusing the information from the previous run. The authors provide a theoretical analysis and empirical results, where they compare against one related baseline and provide extensive results on 3 diverse benchmarks comprising 378 tasks.

**Strengths:**

- Extensive results on 3 diverse tabular benchmarks.

**Weaknesses:**

- **Writing is very unclear** (A few examples out of many):

    **In Section 5:** the manuscript refers to Algorithm 1 and then continues with $S$, $C_0$ without describing them.

    **Algorithm 1**: $(C_k)_k$ is not explained and $k$ is not defined, while additionally it is added twice. Only later in the manuscript, $C_k$ is defined as rungs.

    **Section 7.1:** Hyperband is mentioned, but the plot shows iSHA and PASHA, I am not sure how to understand the sentence.

- While the authors do mention sample and evaluation efficiency in the related work, they do not provide an introduction to methods that combine both. For example, model-based methods that do not adhere to a SHA schedule [1][2] but use a dynamic budget allocation, or methods that sample the fidelities together with the hyperparameter configurations [3].

    **As such, I consider the related work rather incomplete.**
- Only one baseline is included in the experiments.
- I believe the future belongs to methods that do not follow a static schedule, but a dynamic one. Since with a static schedule, even if a hyperparameter configuration were to diverge/stagnate, one would still need to follow the schedule. As such I believe the work will not have an impact in the field.
- Considering the SHA schedule, there are 2 parts, the max budget and the min budget that a configuration will be run to evaluate the performance (the min budget in this case would correspond to the first rung). The authors describe how to increase the max budget, when there is already an existing run, in this example, one could reuse the results from before instead of running everything from scratch. However, what is more important in my perspective, is how to define the $r_{min}$ for the initial run, since that is the fidelity that should be representative of the performance of a hyperparameter configuration.

[1] Wistuba et al. "Supervising the multi-fidelity race of hyperparameter configurations." Advances in Neural Information Processing Systems 35 (2022): 13470-13484.

[2] Kadra et al. "Power Laws for Hyperparameter Optimization." Thirty-seventh Conference on Neural Information Processing Systems (2023)

[3] Kandasamy, Kirthevasan, et al. "Multi-fidelity bayesian optimisation with continuous approximations." International Conference on Machine Learning. PMLR, 2017.

**Questions:**

- **"state-of-the-art algorithm PASHA"**

    Based on what results is PASHA state-of-the-art?
- Could the authors provide a few descriptive statistics on what is the mean improvement and mean degradation for iSHA and PASHA?
- I would recommend the authors to reinforce the related work with the most recent practices regarding multi-fidelity BO.
- I would suggest the authors to update the manuscript and improve readability.
- I would additionally recommend the authors to include more baselines in the experiments.

---

### Official Review · Reviewer_1KJi · 2023-10-28

**Soundness:** 2 fair
**Presentation:** 2 fair
**Contribution:** 2 fair
**Rating:** 3
**Confidence:** 5

**Summary:**

The paper presents a new method called incremental Successive Halving Algorithm (iSHA) to extend an existing hyperparameter optimization process done with Successive Halving (SH). When the expansion factor eta=2, iSHA doubles the budget and creates new brackets by filling the lowest level of the existing bracket with randomly sampled new hyperparameter configurations. It then completes each bracket using the SH algorithm. This allows partial reuse of previous runs, speeding up the process.

The paper also provides theoretical analysis for both ASHA and iSHA. Experiments comparing iSHA to PASHA are done on four different search spaces. Overall, iSHA allows seamlessly continuing an SH hyperparameter optimization run by efficiently reusing previous evaluations.

**Strengths:**

The authors propose an approach to address the tricky issue of selecting hyperparameters for hyperparameter optimization methods, particularly when the choices can strongly impact final performance. Their idea of increasing the R parameter in SH at lower cost could be useful for practitioners.

To build confidence in their method, the authors provide theoretical analysis. They also analyze ASHA in a similar theoretical manner.

**Weaknesses:**

The method to continue SH is relatively straight-forward. Other equally simple methods are not discussed. Just one example: Assume we have 2N completed brackets and we want to increase the budget from R to 2R. What we could do is merge the 2N brackets into N brackets and only run SH for the newly introduced level. Then continue with SH as usual.

The claim that their method outperforms ASHA lacks evidence. The asynchronous issues with ASHA are less relevant given the massive parallelization speedups. In my opinion, the fact that iSHA is synchronous is a strong limitation.

The empirical analysis in the paper focuses only on PASHA, SH, and iSHA. However, it would strengthen the work to include the following additional baselines for comparison:

- ASHA: As the authors mention, ASHA is an important algorithm to include. Its performance compared to PASHA, SH and iSHA should be analyzed. If ASHA was already included and I missed it, please point me to where it is discussed.

- Training top k configurations (for k=1,...): Evaluating performance when simply training the top k configurations found by SH for a larger budget would provide a naive but fast and likely competitive baseline. This would demonstrate the value of more sophisticated methods like iSHA.

- Naive continued SH: An additional baseline could be to continue SH and ignore that previous runs are incomplete. If the configuration with highest val score happens to be among the incomplete ones, just train until completion.


Comparing only accuracy or budget is insufficient - a scatter plot on budget vs performance axes, counting the number of times one method dominates the other, would be better.

Figure 3 is unreadable. It's impossible to quantify dots above or below the 0 line.

Overall, the empirical methodology needs more baselines and better evaluation metrics to demonstrate advantages. In particular, a comparison to ASHA is missing.

**Questions:**

How do you continue configurations? From scratch or from a checkpoint? Given some of the benchmarks, I assume the former.

---

### Official Review · Reviewer_rzKT · 2023-11-01

**Soundness:** 2 fair
**Presentation:** 3 good
**Contribution:** 3 good
**Rating:** 5
**Confidence:** 2

**Summary:**

Most state-of-the-art multi-fidelity methods rely on successive halving as a sub-routine to allocate resources to the evaluation of hyperparameter configurations. The idea is to evaluate a set of configurations for a minimum resource budget, e.g. one epoch, and then to discard the worst half and continue the better half for twice as much budget. This process is iterated until either only a single configuration survives or until some maximum budget is reached.

While very successful in practice, a caveat of successive halving is how to set the minimum and maximum budget before the optimization process starts. For example, setting the maximum budget too small might lead to premature termination of hyperparameter configurations, whereas too large values lead to a high resource consumption. This paper presents a modification of successive halving that allows adapting the maximum budget during optimization, such that a previous run of successive halving is continued without rerunning previous evaluated configurations.

**Strengths:**

- The visualizations in Figure 1 and the pseudo code help a lot to understand the proposed method.

- Overall, I found the paper to be well written and clearly structured.

**Weaknesses:**

- While I personally found the paper easy to follow, uninitiated readers might have some troubles to understand the paper in detail, since it uses a lot of jargon (e.g what means budget for for evaluating a hyperparameter configuration)


- I think the paper needs to better motivate the proposed approach. First, the introduction lists all the relevant hyperparameters of successive halving but the proposed method only adapts the maximum budget. It's not clear why this is more important to adapt than, for instance, the minimum budget. The paper would benefit from discussing this choice.
It would also be helpful if the paper could show some realistic use cases where it is unclear how to set the maximum budget or where a poorly chosen maximum budget leads to severe performance loss. Especially given that most benchmarks in the literature provide a predefined maximum budget, demonstrating scenarios where this causes issues would strengthen the motivation.




- The empirical evaluation in the paper could be strengthened in a few ways:
First, directly comparing the proposed method to ASHA would make the results more convincing, rather than just reporting PASHA outperforms ASHA from the previous work. Reproducing a comparison to ASHA demonstrates good scientific practice.
Second, while the method achieves a reduction in runtime compared to SHA, the decreases are relatively modest at 25% for η=2 and 15% for η=3. Providing additional experiments on more complex tasks/datasets could help show if the benefits of PASHA scale to more difficult optimization problems.

**Questions:**

- Figure 3: Could you also mark the mean or median in these plots?

 - How often is the maximum budget increased? Is it always increased after each bracket, or can it also be kept fixed?


### Typos:
- \eta = 85% I guess it should mean \eta = 3

**Details Of Ethics Concerns:**

No concerns

---

### Official Review · Reviewer_DLma · 2023-11-08

**Soundness:** 2 fair
**Presentation:** 2 fair
**Contribution:** 1 poor
**Rating:** 3
**Confidence:** 5

**Summary:**

The authors propose iterative successive halving (iSHA) as an extension to the successive halving algorithm which extends an original run of SHA to a higher maximum resource by reusing computation of partially trained configurations.  The authors study iSHA and shows in the limit it can achieve 1/\eta savings over SHA where \eta is the promotion rate.   Finally, the authors propose an incremental version of Hyperband which comes with same guarantees as Hyperband.  Experiments comparing iSHA to
SHA and a more resource efficient variant of ASHA called Progressive ASHA (PASHA) shows iSHA to outperform more frequently in terms of speed and selection quality.

**Strengths:**

The primarily strength of this paper is it's a simple and intuitive extension to SHA/Hyperband.  The theoretical analysis of ASHA provides insight in the budget constrained setting but the rate of incorrect promotions for ASHA gets smaller with larger set of configurations unless configurations are drawn adversarially.

**Weaknesses:**

- The speedup of iSHA over SHA is effectively upper-bounded by 1/eta so benefit of the extension is somewhat incremental.
- Experiments are limited to fairly simple surrogate benchmark.  I encourage the authors to evaluate iSHA on more challenging benchmarks like NASBench201 and NASBench301.
- The authors exclude a comparison to ASHA with resumption, which with SHA, are one of the two baselines to beat.

**Questions:**

- What are the mean and standard deviation of iSHA and PASHA on the benchmarks studied?
- How dependent is iSHA on \eta?  How do results look for \eta=4?
- PASHA paper showed much more significant speedups than ASHA on the benchmarks they evaluated.  Why are the speedups in the empirical section of this paper much more limited?

---

### Author Response · Authors · 2023-11-18
**Joint Response to the Reviews**

Thank you for reading our paper and providing helpful and valuable comments. We are pleased that most of you found it easy to read and understand.

Regarding the contribution of our paper, we have the feeling that the theoretical contribution, which is the core of our paper, is not really valued in the expert opinions. We not only analyze our proposed iSHA algorithm quite extensively but also the competitor ASHA.
It is certainly no exaggeration to say that theoretical contributions like ours are rarities in the fields of HPO and AutoML. This aspect alone makes our overall contribution stand out from the typical literature.
However, it is worth emphasizing once again how important it is to provide theoretical guarantees for algorithmic approaches, as this gives more trust in the methods and also makes clear statements about their (mis-)behavior. This is actually good scientific practice.

Moreover, we are the first ones to point out the theoretical limitations of the frequently used algorithm ASHA. Even their argument based on the law of large numbers only helps to a limited extent because the promotion cost of a newly sampled optimal solution also increases over the runtime of ASHA. We are not saying that practitioners should only use iSHA from now on, but we want to raise awareness of the limitations that the asynchronous promotions in ASHA and PASHA bring into play. We showed these limitations to hold in theory and also observed these in practice, i.e., in our empirical evaluation, which is mostly there to demonstrate that our theoretical results also hold when applying the algorithms.

In addition, the asynchronous nature of ASHA might lead to a massive parallelization speedup for NAS when a GPU cluster is available. However, there is also a non-negligible number of practitioners who do not have such a cluster available, but only one GPU. In such a case, the speedup of an asynchronous approach is lost. Moreover, in [2] the authors point out that practitioners, in particular those working with deep learning methods, consider HPO settings with very limited budget as we are considering in our paper. And if, on top of that, one has to deal with the issues we are pointing out here for these approaches, then it is only desirable to have an alternative coming with a provable advantage, as we offer here with iSHA.

We can easily add a comparison to ASHA itself and some more benchmark datasets like NASBench to our experimental results, but please note that PASHA outperformed ASHA anyway on each dataset in the experiments in [1], and in preliminary experiments we found PASHA to dominate ASHA in our setting, yielding better or equal performance spending less budget.
To achieve an optimal tradeoff between performance and parallelism, probably the need for a middle-ground solution arises, e.g., a delay of promotions for instances that are hard to solve during the synchronous evaluation. Nevertheless, this is out of the scope of our paper and can rather be considered as a possible future work.

[1] Ondrej Bohdal, Lukas Balles, Martin Wistuba, Beyza Ermis, Cedric Archambeau, Giovanni Zappella. PASHA: Efficient HPO and NAS with Progressive Resource Allocation. ICLR, 2023.
[2] Mallik, Neeratyoy, et al. "PriorBand: Practical Hyperparameter Optimization in the Age of Deep Learning." arXiv preprint arXiv:2306.12370 (2023), Accepted at NeurIPS 2023 (https://neurips.cc/virtual/2023/poster/70135).

---

> ### Comment · Reviewer_rzKT · 2023-11-22
> **reply**
>
> I thank the authors for their reply. I agree that theoretical guarantees help us to better understand algorithms and that there is arguably a lack in the literature. Indeed the paper has some merits and extends the current literature. I apologize that I didn't sufficiently emphasize this in my review.
>
> However, I don't follow their line of argument regarding the empirical evaluation. If the point of the empirical evaluation is just to raise awareness of the shortcomings of ASHA - which is perfectly valid - then why not include ASHA in the comparison? For the theoretical analysis to be impactful, it would be necessary to show that the underlying assumption, in this case, the asynchronous promotion, is an actual problem.
>
> Second, if practitioners have only access to a single GPU, ASHA is obviously the wrong method in the first place and one ought to use a non-distributed method such as vanilla SHA. However, I am wondering how useful iSHA is in this setting, given that its theoretical improvement over SHA is effectively upper bound by 1/eta.
>
> Apart from this, except for the point about the comparison with ASHA, none of my other points (too much jargon, better motivation, improving figures, question about the maximum budget) have been addressed by the authors. I will therefore stand with my original score.

---

> > ### Comment · Reviewer_nzDd · 2023-11-22
> > **Response to the authors**
> >
> > I thank the authors for their reply. As noted by reviewer rzKT, the authors provided more of a general reply, rather than answer any of the points that I raised. I find the related work lacking. I raised a similar point to reviewer rzKT regarding the priority/importance of adapting the minimal budget compared to the maximal budget, since, when working with multi-fidelity optimization, what is more important is the initial proxy, which should be representative of the performance of a hyperparameter configuration. The paper is additionally not written clearly, while I am familiar with the related work in the domain, the paper uses various terms without defining them before. The experimental section is additionally lacking, featuring only one baseline. Lastly, in my perspective, I believe the proposed method will not have an impact in the domain.
> >
> > Based on the above points and based on the points raised by the other reviewers, I will keep my original score.

---

> ### Comment · Reviewer_DLma · 2023-11-23
> **Post author response**
>
> As acknowledged in my review, I appreciate the theoretical analysis as an important contribution to the paper. However, the weaknesses I brought up were not adequately addressed so I will maintain my score.

---

### Meta-Review · Area_Chair_KwDn · 2023-12-10

**Metareview:**

Reviewers were not convinced about the the novelty of the work presented in the paper and they found the empirical evaluation unconvincing. After the rebuttal, concerns raised by the reviewers remained, in part because the answers provided by the authors were not specific enough. I would encourage the authors to try to clarify the contributions of their work and revise the manuscript to get their point across.

**Justification For Why Not Higher Score:**

None of the reviewers voted for acceptance.

**Justification For Why Not Lower Score:**

N/A

---

### Decision · Program_Chairs · 2024-01-16

Reject